# Warmth Centrality in Social Cognitive Networks of Fairness Reputation Across Players in the Ultimatum and Dictator Games

**DOI:** 10.3390/bs15111537

**Published:** 2025-11-11

**Authors:** Yi Zhao, Yangfan Liu, Ting Xu, Baoming Li, Zhong Yang

**Affiliations:** 1Zhejiang Philosophy and Social Science Laboratory for Research in Early Development and Childcare, Hangzhou Normal University, Hangzhou 311121, China; yyycbb14@163.com (Y.Z.); liuyangfan85@163.com (Y.L.); xuting3@stu.hznu.edu.cn (T.X.); 2Institute of Brain Science, School of Basic Medical Sciences, Hangzhou Normal University, Hangzhou 311121, China; 3Department of Psychology, Jing Hengyi School of Education, Hangzhou Normal University, Hangzhou 311121, China

**Keywords:** social cognition, fairness reputation, warmth–competence model, network analysis, economic games

## Abstract

Fairness reputation refers to the perception of others’ adherence to fair norms based on their behaviors. However, previous studies often rely on simple correlation and regression analyses without comparing cognition across roles in the ultimatum game (UG) and the dictator game (DG). Our study measured the categorical and two-dimensional cognitions (warmth-competence) of participants with different social value orientations toward proposers, responders, and dictators with varying fairness reputations. We found that proposers and dictators with fairness reputations were perceived more positively, and individualists could better distinguish between them. Regarding responders with fairness reputations, they were perceived as more fair, trustworthy, and competent, but less altruistic, cooperative, and warm. The social cognitive network of responders differed from those of proposers and dictators, with warmth cognition being central to three roles, supporting the warmth–competence model. This study highlighted the differential impact of fairness reputation in shaping social cognitions, providing insights into understanding social interactions.

## 1. Introduction

Fairness is a core social norm that underpins honesty, trust, and reciprocity, and is crucial for stable cooperation ([10]; [22]). Individuals prefer cooperating with fair individuals to maintain stable collaborations ([6]; [14]). Thus, understanding whether someone has enforced fairness norms in their past behavior (i.e., fairness reputation) is crucial in social interactions. The Ultimatum Game (UG) and Dictator Game (DG) assess fairness in decision-making ([48]; [63]). In the UG, a proposer offers a split, and a responder can accept or reject it; a rejection results in no earnings ([29]). Fair proposers typically offer 40–60%, while unfair ones offer less than 20–30% ([28]; [39]; [55]). Fair responders reject unfair offers, while unfair responders accept them ([57]). The DG simplifies this by giving the dictator full control, with fairness measured by whether they distribute at least 30% ([33]).

Research has demonstrated that fair proposers and dictators were perceived positively, while unfair ones elicit negative social cognition ([68]). Fair proposers were seen as more trustworthy and likable, whereas unfair ones elicit moral aversion ([19]). Similarly, fair dictators are perceived as more credible and warm ([23]; [33]). However, the underlying motives of fair proposers and dictators may differ ([65]). Proposers offered fair allocations by suppressing self-interest to adhere to fairness rules (altruistic fairness) or strategic considerations to minimize monetary losses from rejection (strategic fairness) ([40]). In contrast, dictators did not fear rejection, so their equitable distributions were usually viewed as acts of pure selfless fairness ([65]). Thus, there may have been differences in the social cognition of proposers and dictators, with or without fairness reputations that remain to be clarified.

Research on how responders’ fairness reputations affect social cognition has yielded mixed results. Some argue that rejecting unfair offers enforces fairness norms and earns moral approval, while acceptance reflects self-interest ([4]; [43]). Others suggest that rejection is an intuitive but disruptive act, whereas acceptance is seen as a rational choice that maintains cooperation, leading to perceptions of warmth and likability ([38]; [53]; [67]). These inconsistencies highlight the need for more advanced methods to assess social cognition.

Most research has used categorical ratings to measure social cognition, but the warmth–competence model could be a more integrative approach. Categorical ratings like trustworthiness and likability failed to capture the nuances across fairness reputation roles ([19]; [38]). Researchers have suggested that warmth and competence are two universal dimensions of social cognition ([5]; [25]). Warmth referred to the perceived intentions, and competence referred to the perceived ability to execute those intentions ([25]). Studies show that unfair responders appeared warmer, while fair dictators were warmer than those who were unfair ([33]; [34]; [38]). However, the relationship between warmth, competence, and categorical social cognition remains unclear ([53]). Warmth is widely recognized as a primary dimension in social judgment, providing a basis for evaluating potential threats or benefits in social interactions and for inferring others’ intentions and moral motives ([25]). Compared with warmth, competence is often considered secondary, as it is related more to ability than to social intention. Observers have inferred moral character from others’ behaviors, and such inferences critically shaped subsequent moral evaluations ([51]). In fairness-related contexts, fairness behavior conveys moral intention and adherence to social norms ([20]; [51]), suggesting that one’s fairness reputation is primarily processed through the warmth dimension.

Network analysis maps relationships between multiple variables and can reveal links between categorical social cognition and warmth–competence dimensions ([3]). Compared to traditional statistical methods (e.g., regression analysis), network analysis allows for controlled variable comparisons, identifies central cognition via node attributes, and mitigates overfitting using LASSO (Least Absolute Shrinkage and Selection Operator) regularization ([7]; [17]). Thus, we used network analysis to compare the structure of social cognitive networks ([61]).

In addition, social value orientations (SVO) reflect individuals’ preferences for balancing self-interest and others’ welfare ([42]). Prosocial individuals favored cooperation for mutual benefit, while individualists prioritized personal gains ([58]). This may stem from prosocials integrating moral principles into their identity, whereas individualists associate positive emotions with self-serving actions ([37]). Since people tend to project their own motives onto others ([12]), we assessed SVOs, predicting that they would shape observers’ cognition of fairness reputation.

To resolve the above issue, we recruited participants with different SVOs (prosocial and individualistic) and asked them to observe the decision-making behaviors of three roles (UG proposer, UG responder, and DG dictator) as well as to evaluate the categorical social cognition of each role (fairness, trustworthiness, altruism, and cooperation) and the warmth–competence dimensions. We employed network analysis to identify the differences in social cognition across roles under different fairness reputations. The hypotheses were as follows: (1) Roles with a fairness reputation were perceived more positively than those without; (2) dictators with versus without fairness reputations would exhibit a greater divergence in social cognition than would proposers, as their fair behavior is attributed to altruistic rather than strategic motives; (3) the social cognition of responders would be ambivalent, with both positive and negative cognitions held toward fair responders; (4) individualistic individuals would better distinguish fairness reputations across roles than prosocial individuals; and (5) warmth would function as a central node in the social cognitive network, which is closely linked to other categorical cognitions.

## 2. Materials and Methods

### 2.1. Participants

Non-psychology and non-economics majors at a university in Hangzhou were recruited as participants through the university’s bulletin board system. All the participants reported that they had not participated in a similar experiment before. The optimal sample size was estimated using G*Power 3.1 in F-tests (ANOVA, repeated measures) covering between-subject factors, within-subject factors, and within–between interactions. Assuming α = 0.05, power (1 − β) = 0.90, and a medium effect size f = 0.20, the sample size was determined to be 108, which was the largest among all F-tests to detect both the main effects and interaction. The effect size of f = 0.25 was chosen because it represents a medium effect according to [9] ([9]) and has been reported in previous studies examining fairness and social value orientation effects in economic decision-making tasks ([32]). Details of the sample size calculation are attached to the Appendix A. A total of 170 participants took part in this experiment. After examining the data and balancing the number of participants with different SVOs (see Appendix A for details), 122 valid participants were included in the data analyses (61 males ranging from 18 to 29 years; Mage = 20.68 years, *SE* = 1.94). All participants were right-handed, had normal or corrected-to-normal vision, and reported no prior history of neurological or psychiatric disorders. They voluntarily participated in the experiment and signed an informed consent form before the experiment began. After the experiment, they were given a reward of 20 yuan. This study received ethical approval from an institutional research ethics committee prior to data collection.

### 2.2. Stimuli and Procedures

This study utilized a structured survey method (see Figure 1).

After introducing the rules of the decision-making task, a comprehension check was conducted to ensure participants correctly understood the rules. Then, the participants engaged in decision-making processes as players in the corresponding roles, with the aim of enhancing their comprehension of the established rules.

During the observation phase, the participants were asked to observe the decision-making behaviors of other players to learn about their fairness reputations. A player was classified as “fair” if ≥80% of their observed actions met the fairness rule, and they were classified as “unfair” if ≥80% of their observed actions met the unfairness rule. Based on the roles and reputation types, the players were categorized into six types: fair proposer in UG (FP), unfair proposer in UG (UP), fair responder in UG (FR), unfair responder in UG (UR), fair dictator in DG (FD), and unfair dictator in DG (UD). Each identity participated in ten rounds of decision-making.

The proposer’s decision-making procedure is shown in Figure 2A. At the beginning of each trial, the serial number of the current trial was presented for 500 ms. Next, an offer of the proposer for the current trial was displayed on the screen, with “Accept” and “Reject” buttons at the bottom to indicate that the responder’s decision was awaited, lasting 1500 ms. To enhance the sense of reality, two of the ten rounds of decision-making were set to be opposite to the reputation type, serving as fillers. Fair proposers often proposed fair offers as follows (proposer/responder): 5:5, 6:4, 6:4, 5:5, 9:1, 5:5, 8:2, 6:4, 5:5, and 6:4. In contrast, unfair proposers often proposed unfair offers as follows (proposer/responder): 8:2, 5:5, 8:2, 9:1, 9:1, 8:2, 6:4, 9:1, 8:2, and 9:1.

The responder’s decision-making procedure is shown in Figure 2B. At the beginning of each trial, the serial number of the current trial was presented for 500 ms. Next, an offer from the proposer for the current trial was displayed on the screen, with the “Accept” and “Reject” buttons at the bottom to indicate that the responder’s decision was awaited, lasting 1500 ms. Then, the “Accept” or “Reject” button was framed by a green square, indicating the responder’s decision in this trial, lasting 2000 ms. Finally, the outcome of both players in this trial was displayed for 1000 ms. To enhance the sense of reality, two of the ten rounds of decision-making were set to be opposite to the reputation type. Responders with fair reputations often rejected unfair offers as follows (proposer/responder, decision behavior): 8:2 (reject), 6:4 (accept), 9:1 (reject), 8:2 (reject), 9:1 (reject), 9:1 (reject), 8:2 (accept), 8:2 (reject), 5:5 (accept), and 9:1 (reject). In contrast, responders with unfair reputations often accepted unfair offers as follows (proposer/responder, decision behavior): 8:2 (accept), 9:1 (accept), 6:4 (accept), 8:2 (accept), 9:1 (accept), 9:1 (accept), 8:2 (accept), 9:1 (reject), 5:5 (accept), and 8:2 (accept).

The dictator’s decision-making procedure is shown in Figure 2C. The process was similar to the proposer’s except that there was no button at the bottom.

After the observation phase, the participants were required to answer manipulation check questions to assess whether they had learned about the players’ fairness reputations. Afterward, the participants were asked to evaluate their social perceptions of the players’ behavior on a seven-point Likert scale that ranged from 1 (totally disagree) to 7 (totally agree). The evaluations included both categorical social cognition (fairness, trustworthiness, altruism, and cooperation); two dimensions of social cognition, such as warmth (sincere, good natured, warm, tolerant, and friendly); and competence (competent, confident, independent, intelligent, and competitive) ([1]; [25]; [41]). Finally, the participants completed the SVO Slider Measure and demographic questions.

The trial order within each role was fixed rather than randomized to ensure that the participants could form a stable impression of each role’s behavioral pattern across the trials. However, the reputation conditions were counterbalanced between participants: half of the participants first observed roles without a fairness reputation and then those with a fairness reputation, while the other half experienced the opposite order.

The details of comprehension check questions are provided in the Appendix A.

### 2.3. Data Analyses

Firstly, we conducted fairness reputation manipulation checks separately for roles to ensure that participants learned about the fairness reputation of the different players. Paired-samples *t*-tests were conducted on the estimated means of the offers per trial proposed by proposers and dictators under different fairness reputation conditions. A chi-square test was conducted on the predictions of responders’ subsequent choices under different fairness reputation conditions.

Secondly, we compared the categorical social cognition and the two dimensions of the social cognition of prosocial and individualistic participants towards different roles under different fairness reputation conditions. For the proposers and dictators, a series of 2 (fairness reputation: fair vs. unfair) × 2 (SVO: prosocial vs. individualistic) × 2 (role: proposer vs. dictator) mixed-design ANOVAs was conducted. For the responders, a series of 2 (fairness reputation: fair vs. unfair) × 2 (social value orientation: prosocial vs. individualistic) mixed-design ANOVAs was conducted. For the sensitivity power analysis, N = 122, α = 0.05, power (1 − β) = 0.90, detected minimum effect sizes of f = 0.13 for the 2 × 2 × 2 mixed ANOVAs, and f = 0.16 for the 2 × 2 mixed ANOVAs. The above details of the sensitivity power analysis are explicitly reported in the Appendix A. All reported significant effects exceeded this threshold, suggesting they were sufficiently powered. However, we acknowledge that this study may be underpowered to detect very small interaction effects. Where applicable, we used the Welch/Kenward–Roger corrections via the *afex* package (version 1.3-1) in R to ensure robust inference in the presence of potential heteroskedasticity. To control for type I error inflation due to multiple correlated dependent variables, we applied Holm–Bonferroni corrections within the logical hypothesis families. For each experimental hypothesis (e.g., the main effects of fairness reputation, SVO, role, and their interactions), corrections were applied across the six dependent variables (fairness, trustworthiness, altruism, cooperation, warmth, and competence). All reported *p*-values reflect these conservative corrections.

Finally, we examined the relationship between categorical and two-dimensional social cognitions within and between roles. (1) We estimated the structure of the social cognitive networks of proposers, responders, and dictators using the six cognitions as nodes. After applying a nonparanormal (npn) transformation to reduce non-normality, we computed Pearson correlations on the transformed data and estimated the regularized partial-correlation networks separately for each role using the GLASSO (Graphical LASSO) model based on the Extended Bayesian Information Criterion (EBIC) with the default tuning parameter γ = 0.5. Node positions were then optimized for visualization. To assess robustness, we conducted supplementary analyses comparing Pearson (on npn-transformed data) versus polychoric correlations and EBIC γ values of 0.25 and 0.75; these robustness checks are reported in Appendix A. ([24]; [27]). (2) We calculated the centrality indices of the networks, including strength, betweenness, closeness, and expected influence (EI) ([3]; [49]; [60]). (3) We utilized the Network Comparison Test (NCT) version 2.2.2 to compare the social cognitive networks across roles ([61]).

We conducted data analyses using SPSS version 26.0, R version 4.3.2, qgraph (version 1.9.8), and bootnet (version 1.6), with a statistical significance of *α* = 0.05 (two-tailed). To assess the assumption of homogeneity of variances in ANOVA analyses, we conducted Levene’s tests. In cases where this assumption was violated, we applied the Greenhouse–Geisser correction and reported the adjusted results. All the results were corrected using Bonferroni tests if there were more than three groups in the post hoc and simple effects analyses. All results of the Network Comparison Test (NCT) were adjusted using the Holm–Bonferroni correction to control for multiple comparison errors, with the significance level set at *p* < 0.05.

## 3. Results

### 3.1. Fairness Reputations Manipulation Check

Manipulation checks confirmed successful fairness reputation assignments for proposers, responders, and dictators. The participants’ estimated offers closely matched the predefined values for proposers and dictators, and responders’ expectations aligned with the intended fairness conditions (Figure 3). The specific statistical values, including *t*, *χ*^2^, means, and standard errors, are provided in the Appendix A.

### 3.2. Social Cognition of Roles Under Varying Fairness Reputation Conditions

#### 3.2.1. Homogeneity Test of Variance on Social Cognition of Proposers and Dictators

A homogeneity test of variance was conducted on the warmth ratings for proposers and dictators. The results indicated significant differences in variance between fair and unfair roles across both prosocial and individualistic participants, with a greater variability observed for fair proposers and dictators. However, no significant differences in group covariance matrices were found for other social cognitions (Appendix A for details).

#### 3.2.2. ANOVA Results on Social Cognition of Proposers and Dictators

The means and standard errors (SE) on social cognitions of “fair” vs. “unfair” proposers/dictators are shown in Table 1. The specific statistical values for all ANOVAs, including the *F*, *η_p_*^2^, and Confidence Interval (CI), are shown in Table 2.

Regarding the main effects, we found significant effects of the fairness reputation, indicating that fair players were perceived as fairer, more trustworthy, altruistic, cooperative, warm, and competent than unfair players. Similarly, we found significant main effects of the role, indicating that proposers were perceived as less fair, trustworthy, altruistic, cooperative, warm, and competent than dictators (Figure 4).

Regarding the two-way interactions, we found significant interactions between fairness reputation and role. The simple effect analysis revealed that fair proposers were perceived as more trustworthy, altruistic, cooperative, warm, and competent than unfair proposers; fair dictators were also perceived as more trustworthy, altruistic, cooperative, and warm than unfair dictators, with a larger difference compared to proposers. Additionally, the interactions between role and SVO were also significant. The simple effect analysis indicated that for prosocial participants, proposers were perceived as less fair than dictators; and for individualistic participants, proposers were also perceived as less fair than dictators, with a larger difference than that for prosocial participants (Figure 4).

The specific statistical values for all ANOVAs, including *F*, *η_p_*^2^, means, and standard errors, are provided in the Appendix A.

#### 3.2.3. Homogeneity Test of Variance on Social Cognition of Responders

A homogeneity test of variance was conducted on fairness and competence ratings for responders. The variance in fairness ratings differed by responder fairness among prosocial participants, while no significant variance differences were found for competence ratings or other social cognition dimensions (Appendix A).

#### 3.2.4. ANOVA on Social Cognition of Responders’ Fairness Reputations

The means and standard errors (SE) of the social cognitions of “fair” vs. “unfair” responders are shown in Table 3. The specific statistical values for all ANOVAs, including the *F*, *η_p_*^2^, and Confidence Interval (CI), are shown in Table 4.

We found significant effects of the fairness reputation, indicating that fair responders were perceived as fairer, more trustworthy, and competent than unfair players, but were considered less altruistic, cooperative, and warm than unfair players (Figure 5).

The specific statistical values for all ANOVAs, including the *F*, *η_p_*^2^, means, and standard errors, are provided in the Appendix A.

### 3.3. Relationships Between Categorical and Two Dimensions of Social Cognition

We investigated the relationship between social cognition within and between roles using a regularized partial-correlation network, where the nodes representing warmth, competence, and categorical social cognition were generally interconnected. For the proposer role, warmth was positively correlated with fairness (edge value = 0.33), with trustworthiness (edge value = 0.30), and with cooperation (edge value = 0.45), while competence was positively correlated with trustworthiness (edge value = 0.21) (Figure 6A). For the responder role, warmth was positively correlated with trustworthiness (edge value = 0.35), with cooperation (edge value = 0.35), and with altruism (edge value = 0.38), while competence was positively correlated with fairness (edge value = 0.39) and with trustworthiness (edge value = 0.22). Additionally, there were negative correlations between warmth and fairness (edge value = −0.07), and between competence and altruism (edge value = −0.11) (Figure 6C). For the dictator role, warmth was positively correlated with fairness (edge value = 0.33), with trustworthiness (edge value = 0.42), and with cooperation (edge value = 0.37), while competence was positively correlated with trustworthiness (edge value = 0.11) (Figure 6E).

The results of centrality indices revealed that warmth emerged as the most central node in the networks for the proposer, responder, and dictator, as indicated by high values in strength, closeness, betweenness, and EI, followed by trustworthiness (Figure 6B,D,F). Additionally, cooperation exhibited high centrality within the social cognitive networks for the proposer and dictator roles (Figure 6B,F). The centrality indices metrics for each node within each network are detailed in Appendix A.

The network stability analysis showed adequate to good centrality stability (CS) across all networks (Proposer: Strength = 0.75, Expected Influence = 0.75, Closeness = 0.516; Responder: Strength = 0.283, Expected Influence = 0.594, Closeness = 0.439; Dictator: Strength = 0.75, Expected Influence = 0.75, Closeness = 0.672; all > 0.25; [17]). Detailed bootstrapped difference plots for nodes and edges are provided in Appendix A.

Next, we conducted pairwise comparisons of the social cognitive networks between the proposer, responder, and dictator using the Network Comparison Test (NCT).

For the social cognitive networks of the proposer and responder, there was no significant difference in the global strength of the networks (strength difference = 0.46; *p* = 0.149); however, there was a significant difference in the overall structure of the networks (maximum edge weight difference = 0.39; *p* = 0.002). An examination of individual edges revealed that five edges showed a statistical difference between these two networks (*ps* < 0.05). Compared to the proposer’s network, the responder’s network exhibited weaker associations between warmth and fairness, and between fairness and cooperation. It also showed stronger associations between warmth and altruism, and between competence and fairness, along with an inverse negative association between fairness and altruism.

For the social cognitive networks of the responder and dictator, there was no significant difference in the network global strength of the networks (strength difference = 0.46; *p* = 0.225); however, there was a significant difference in the overall network structure (maximum difference in edge weights = 0.39; *p* = 0.002). Examination of individual edges revealed that four edges showed a statistical difference between these two networks (*ps* < 0.05). Compared to the proposer’s network, the responder’s network exhibited weaker associations between warmth and fairness, and between fairness and cooperation. It also showed stronger associations between competence and fairness, along with an inverse negative association between fairness and altruism.

Regarding the social cognitive networks of the proposer and dictator, no significant differences were found in the network global strength (strength difference < 0.001; *p* = 0.997) or overall network structure (maximum difference in edge weights = 0.12; *p* = 0.916). Details of the edge weights in the social cognitive networks between the three roles are presented in Appendix A.

Furthermore, we compared the social cognitive networks between prosocial and individualistic participants to examine whether different SVO orientations influenced the structure of fairness-related social cognition. The two groups showed generally consistent overall structures and global strengths across the proposer, responder, and dictator networks, with no significant group differences observed, as shown in Appendix A.

## 4. Discussion

This study examined the influence of the fairness reputations of the UG proposer, UG responder, and DG dictator on the social cognition of observers with different SVOs. First, proposers and dictators with fairness reputations were perceived more positively than those without, and the differences in social cognition between dictators with varying fairness reputations were more pronounced than between proposers. Moreover, individualists exhibited greater differences in social cognition of the proposers and dictators, with varying fairness reputations compared to the prosocials. Second, responders with fairness reputations were perceived as more fair, trustworthy, and competent, but less altruistic, cooperative, and warm. Finally, warmth functioned as a central node in the social cognitive network of the proposer, responder, and dictator, though the social cognitive network of the responder was significantly different from the other two roles. Our findings elucidated the different impact of fairness reputation on social cognition across roles, expanded the warmth–competence model, and highlighted the significance of fairness in social interactions.

### 4.1. The Influence of Fairness Reputation on Social Cognition to Proposers and Dictators

We found that proposers and dictators with fairness reputations tended to be perceived as more fair, trustworthy, altruistic, cooperative, warm, and competent. Their fair offers may have reflected adherence to the fairness norm ([22]) and aligned with the cooperation norm of maximizing public benefits ([31]; [67]). The consistent and predictable behavioral patterns served as reliable signals and enhanced trustworthiness ([19]; [21]). Moreover, their ability to process moral information and self-regulate may have been interpreted as being aligned with moral intelligence theory, which could partly explain their perceived competence ([56]). Notably, warmth perceptions exhibited greater heterogeneity, being higher when fair behavior was attributed to prosocial motivations such as fairness or altruism, and being lower when attributed to strategic considerations or self-interest ([15]; [44]).

In addition, the difference in social cognition between dictators with and without fairness reputations was more pronounced than between proposers, possibly due to the differing motivations of these two roles. The fairness behavior of the proposer in the UG may be driven by strategic motivation to avoid rejection and maximize self-interest rather than pure altruism ([36]; [62]). In contrast, dictators in the DG may be perceived as driven by fairness norms or altruistic motivations, given their absolute power in determining outcomes for both participants ([26]). Therefore, the social cognition between dictators with and without fairness reputations was more polarized than between proposers.

### 4.2. The Influence of Fairness Reputation on Social Cognition of Responders

On the one hand, responders with fairness reputations tended to be perceived more positively, being seen as somewhat fairer, more trustworthy, and more competent compared to those without fairness reputations. By consistently rejecting unfair offers and sacrificing self-interest to enforce fairness norms, these responders may have strengthened observers’ fairness perceptions ([22]). Moreover, the behavioral consistency of responders with fairness reputations appeared to be interpreted as a stable and reliable signal, associated with higher perceived trustworthiness, which is consistent with proposers and dictators with fairness reputations ([19]; [21]). This result aligned with previous research, which also found a trend of higher competence perceptions toward responders with fair reputations ([38]). In our study, participants observed both responders with and without fairness reputations, leading to a comparison between the two responders, which made the differences more pronounced ([34]).

On the other hand, responders with fairness reputations may also be perceived more negatively, being seen as less altruistic, cooperative, and warm. Their rejections harmed the proposer’s interests and minimized the total earnings ([29]), violating the cooperation norm of maximizing collective benefits and contradicting the essence of altruism, which aims to increase the welfare of others ([31]; [67]). Moreover, rejection behaviors were attributed to aroused anger and seen as spiteful actions, which were contrary to the definition of warmth, leading to them being perceived as lacking warmth ([25]; [30]; [38]).

### 4.3. Relationship Between Categorical and Two Dimensions of Social Cognition

Warmth emerged as the central node across all roles’ social cognitive networks, validating its primacy in the warmth–competence model ([25]; [66]). The prioritization of warmth judgments is rooted in the evolutionary necessity of rapidly discerning others’ intentions as friendly or hostile, thereby providing a basis for decisions regarding potential threats ([11]). For proposers and dictators, warmth was more closely linked to fairness, trustworthiness, and cooperation, as their fair behaviors directly reflected the connection between reputation and benefits ([18]; [34]; [45]). However, the weak association between warmth and altruism suggested the motive attributions were strategic (e.g., reputation maintenance) rather than altruistic ([36]). For responders, warmth was more closely linked to trustworthiness, cooperation, and altruism, as rejection behaviors served as altruistic punishments but undermined cooperation, preventing partners from gaining benefits ([8]).

Competence showed only a weak correlation with trustworthiness across all roles’ social cognitive networks, as it is regarded as an “instrumental trait” that only concerns task efficiency. Low competence may undermine the joint goal and reduce trustworthiness, while high competence may increase the perception of “strategic exploitation,” thereby diminishing trust. Therefore, although competence can predict trustworthiness to some extent, warmth is still needed to help judge intentions ([11]; [35]). In addition, the competence perceptions of responders were correlated with fairness. This may be because the rejection behavior of responders is a costly punishment, where they sacrifice their own interests to enforce fairness norms and achieve moral goals ([4]; [64]).

### 4.4. The Influence of Observer Social Value Orientation on Social Cognition of Fairness Reputation

Our study found some evidence that individualists and prosocials may differ in their perception of fairness between proposers and dictators; however, this pattern was observed specifically for fairness ratings after applying rigorous statistical corrections. One possible explanation for this pattern is that observers tend to assume that others share similar motivations, which may lead to differences in their perception and evaluation of others’ behaviors ([16]). Under this account, prosocials may be more inclined to assume that others also have prosocial motivations, while individualists may tend to assume that others are motivated by self-interest ([2]; [47]). This could lead individualists to infer that proposers’ fair distributions represent strategic behaviors to avoid rejection, while perceiving dictators’ fair distributions as potentially driven by genuine altruism. However, we stress that this interpretation remains speculative, as motive attributions were not directly measured in the present study. Future research should therefore incorporate explicit assessments of motive attributions or use modeling approaches to test whether reputation evaluations are indeed mediated by perceived intentions.

Notably, prosocials showed lower consistency in their fairness perceptions of responders with fairness reputations, as prosocials have distinct motivations of inequity aversion and joint gain maximization ([42]). Individuals with inequity aversion were sensitive to the equality between payoffs, and the rejection behaviors made the outcomes more equal, increasing observers’ fairness perceptions. In contrast, those focused on joint gain maximization perceived rejection as undermining overall gains, reducing observers’ perceptions of fairness.

### 4.5. Limitations and Future Perspectives

First, the experimental design restricted fairness reputation learning to observing distributional behaviors in classic UG/DG paradigms, excluding multimodal cues like verbal/prosodic signals ([50]; [59]), potentially compromising its ecological validity. Future studies could utilize virtual reality to integrate visual–semantic–prosodic information ([46]) and adopt real-world scenarios ([33]) to improve the design’s ecological and external validity. Additionally, our sample consisted solely of Chinese university students, which limits the generalizability of the findings across cultural contexts. Notably, the competent-cold pattern observed in fair responders in our study has also been documented in international and cross-cultural data, as well as in research on individual stereotypes ([13]). Future research could examine whether this competent-cold pattern for fair responders is observed across different cultural contexts.

Second, although we found differences in social cognition toward proposers and dictators from the perspective of group means, there may be individual differences in motive attribution (strategic vs. altruistic fairness) ([65]). Future research should directly measure motive inferences per trial and employ neurophysiological methods to examine neural correlates of these differences—particularly the ventrolateral prefrontal cortex in negative reputation processing and medial prefrontal cortex in context-dependent valuation when choosing partners ([38]; [54]). Moreover, although the deterministic behavioral sequences and filler trials followed established paradigms in social decision-making research to ensure consistent reputation cues, such fixed patterns may have increased the salience of fairness cues and reduced ecological validity. Future studies should employ more dynamic, probabilistic, and interactive designs to better approximate real-world reputation formation.

Third, a potential limitation of the present study concerns the post hoc balancing of SVO groups. To achieve equal group sizes and enhance the internal validity of between-group comparisons, we included only prosocial participants with relatively larger SVO angles. Although this procedure helped establish clearer group boundaries and improved the interpretability of the SVO-based contrasts, it also reduced the external validity of the findings. The final balanced sample no longer reflects the natural distribution of SVO types in the broader population, where prosocial orientations are typically more prevalent. Therefore, the generalization of the present results to populations with different SVO compositions should be made with caution.

Fourth, we included a small number of inconsistent trials to enhance the ecological validity of the task and to prevent participants from perceiving the fairness reputation manipulation as overly artificial or deterministic. However, because these fillers intentionally deviated from the assigned fairness reputation, they may have introduced ambiguity regarding the strength or consistency of each identity’s reputation. Although the presentation order was fixed within each role and the assignment of fair and unfair reputations was counterbalanced across participants, the present design does not allow us to determine whether the inclusion of fillers attenuated or polarized participants’ fairness judgments. Future studies could employ stimulus materials without such fillers to establish a more distinct and uncontaminated manipulation of fairness reputation and to assess the potential impact of inconsistent behavioral cues on reputation perception.

Finally, this study primarily focused on the impact of fairness reputation on social cognition; however, reputation may also influence decision-making in real situations. For example, individuals were more likely to engage in proactive behaviors when they perceived others as high in warmth, such as actively helping, and reactive behaviors when they perceived others as high in competence, such as accepting advice ([52]). Future research could use behavioral economic tasks (e.g., trust games or the prisoner’s dilemma) to examine whether observers engage in more trusting and cooperative behaviors with fair proposers and dictators. Furthermore, conflicting social cognition of responders may predict increased investment but less cooperation. Given that the present paradigms (the UG and DG) inherently involve fairness, altruism, and reciprocal cooperation, participants’ perceptions and evaluations may have been influenced by multiple overlapping social norms. Future studies could further disentangle these constructs by employing paradigms that isolate fairness from other cooperative motives and by testing whether similar cognitive patterns of fairness reputation emerge.

## 5. Conclusions

To conclude, this study systematically revealed the influence of fairness reputation on social cognition of different roles (proposers, responders, and dictators). First, fairness reputation reinforced positive trait inferences for proposers and dictators, but it induced contradictory cognition of ‘high competence/low warmth’ for responders. Secondly, individualists were better able than prosocial individuals to differentiate fairness reputation between proposers and dictators, which may stem from motivational differences in attributions between the two. Additionally, the results of the network analyses showed that the social cognitive networks of responders were significantly different from those of the proposers and dictators; however, warmth perception served as the central hub of the social cognitive networks in all three networks, highlighting that warmth cognition occupies an important position in the cognition of fairness reputation. Based on these findings, this study not only shed light on the important role of fairness reputation in social cognition, but it also supported the warmth–competence model and explored its relationship with categorical social cognition.

## Figures and Tables

**Figure 1 behavsci-15-01537-f001:**
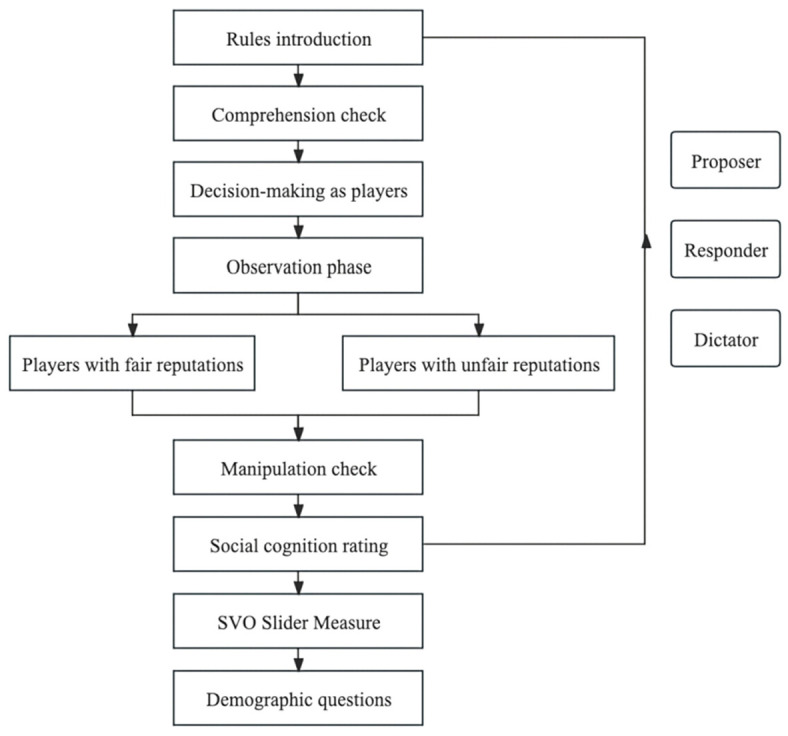
The survey processes.

**Figure 2 behavsci-15-01537-f002:**
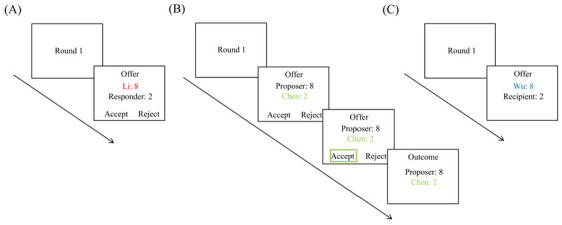
Illustration of the observation phase: (**A**) proposer, (**B**) responder, and (**C**) dictator.

**Figure 3 behavsci-15-01537-f003:**
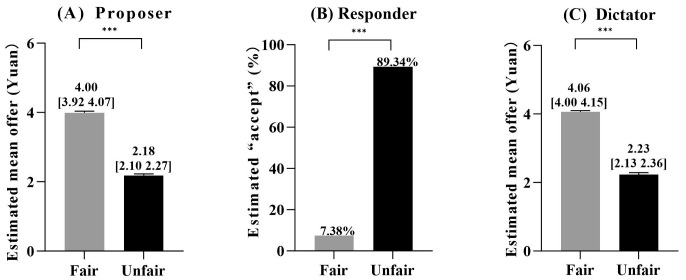
The results of the fairness reputation manipulation checks: (**A**) proposer role, (**B**) responder role, and (**C**) dictator role. Significance level: *** *p* < 0.001. Error bars indicate standard errors.

**Figure 4 behavsci-15-01537-f004:**
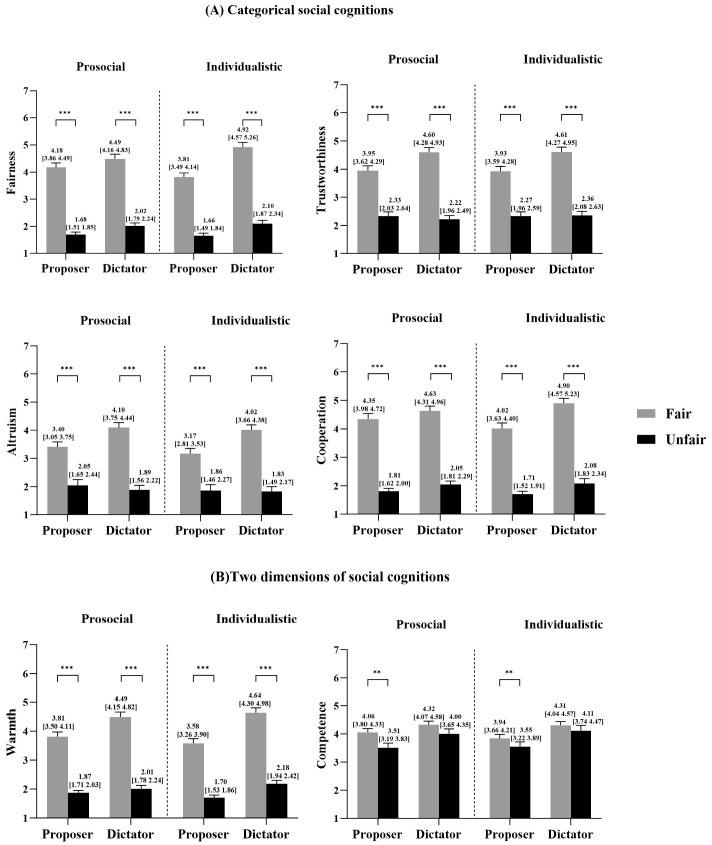
Social cognition of proposers and dictators under different fairness reputation conditions. (**A**) Categorical social cognition ratings: fairness, trustworthiness, altruism, and cooperation; (**B**) two dimensions of social cognition ratings: warmth and competence. Significance level: ** *p* < 0.01; *** *p* < 0.001. Error lines indicate standard errors. Note. FP = fair proposer; UP = unfair proposer; FD = fair dictator; UD = unfair dictator.

**Figure 5 behavsci-15-01537-f005:**
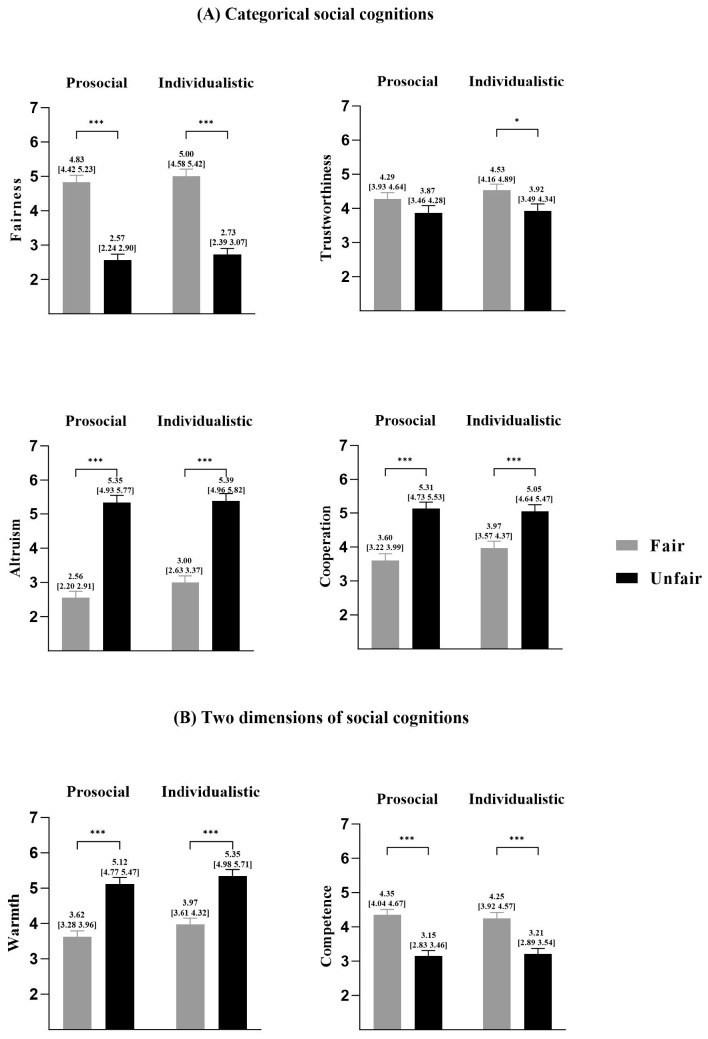
Social cognition of responders under different fairness reputation conditions. (**A**) Categorical social cognition ratings: fairness, trustworthiness, altruism, and cooperation; (**B**) two dimensions of social cognition ratings: warmth and competence. Significance level: * *p* < 0.05; *** *p* < 0.001. Error lines indicate standard errors. Note. FR = fair responder; UR = unfair responder.

**Figure 6 behavsci-15-01537-f006:**
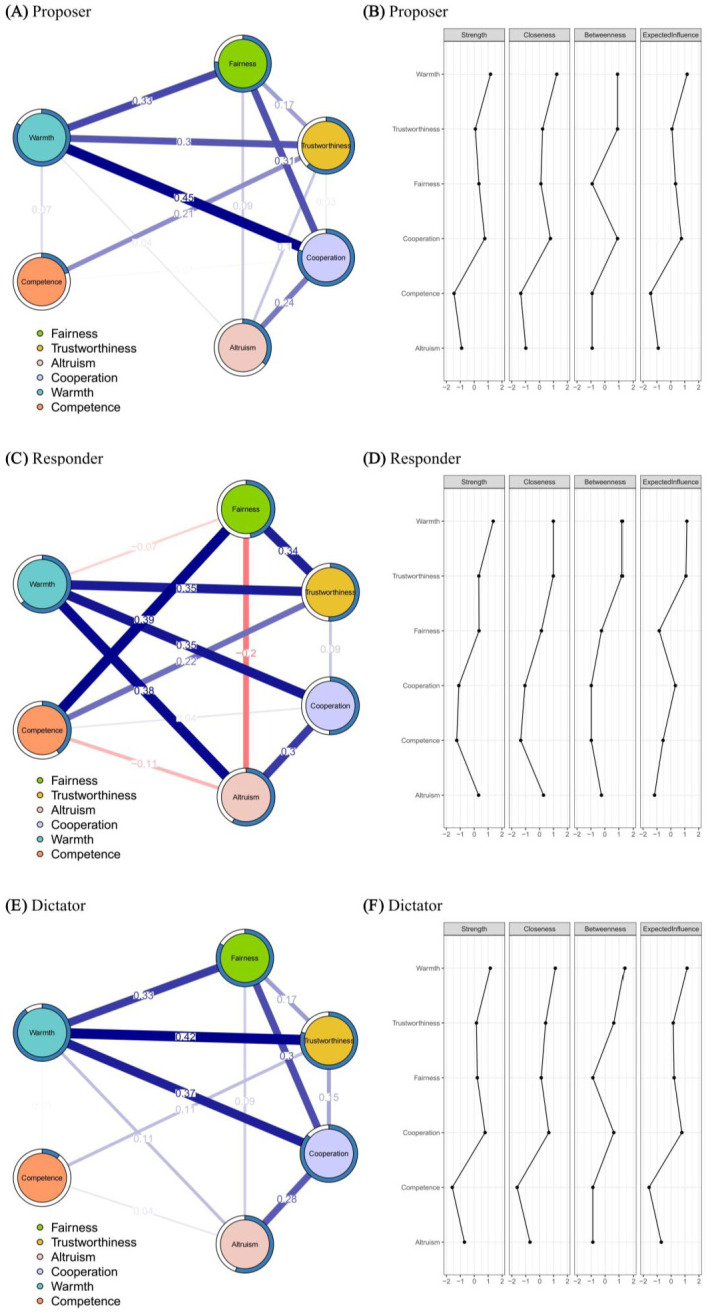
Social cognitive network of (**A**) the proposers and their (**B**) standardized centrality estimates; (**C**) the responders and their (**D**) standardized centrality estimates; and (**E**) the dictators and their (**F**) standardized centrality estimates. The nodes denote categorical and two dimensions of social cognition. Regularized partial correlations were estimated using the EBIC-GLASSO model based on Pearson correlations after nonparanormal (npn) transformation. The pie chart surrounding each node represents its predictability (R^2^), with a larger proportion of blue indicating a higher explained variance. The pie chart surrounding a node illustrates that node’s predictability within the network, with a larger proportion of blue indicating higher predictability. The edges indicate regularized partial correlations between two nodes. The edge values represent the strength of partial correlations; a larger value indicates a stronger association between two nodes and a thicker edge. The color of the edge represents the direction of the correlation, with blue edges indicating positive associations and red edges indicating negative associations.

**Table 1 behavsci-15-01537-t001:** The means and standard errors (SE) on social cognitions of “fair” vs. “unfair” proposers/dictators.

Social Cognition	SVO	Role	Fairness Reputation	Mean (SE)	CI
Fairness	Prosocial	Proposer	Fair	4.17 (0.16)	[3.86, 4.49]
			Unfair	1.68 (0.09)	[1.51, 1.85]
		Dictator	Fair	4.49 (0.17)	[4.16, 4.83]
			Unfair	2.02 (0.11)	[1.79, 2.24]
	Individualistic	Proposer	Fair	3.81 (0.16)	[3.49, 4.14]
			Unfair	1.66 (0.09)	[1.49, 1.84]
		Dictator	Fair	4.92 (0.17)	[4.57, 5.26]
			Unfair	2.10 (0.12)	[1.87, 2.34]
Trustworthiness	Prosocial	Proposer	Fair	3.95 (0.17)	[3.62, 4.29]
			Unfair	2.33 (0.15)	[2.03, 2.64]
		Dictator	Fair	4.60 (0.16)	[4.28, 4.93]
			Unfair	2.22 (0.13)	[1.96, 2.49]
	Individualistic	Proposer	Fair	3.93 (0.17)	[3.59, 4.28]
			Unfair	2.27 (0.16)	[1.96, 2.59]
		Dictator	Fair	4.61 (0.17)	[4.27, 4.95]
			Unfair	2.36 (0.14)	[2.08, 2.63]
Altruism	Prosocial	Proposer	Fair	3.40 (0.18)	[3.04, 3.75]
			Unfair	2.05 (0.20)	[1.65, 2.44]
		Dictator	Fair	4.10 (0.18)	[3.75, 4.44]
			Unfair	1.89 (0.16)	[1.56, 2.21]
	Individualistic	Proposer	Fair	3.17 (0.18)	[2.81, 3.53]
			Unfair	1.86 (0.21)	[1.46, 2.27]
		Dictator	Fair	4.02 (0.18)	[3.66, 4.38]
			Unfair	1.83 (0.17)	[1.49, 2.17]
Cooperation	Prosocial	Proposer	Fair	4.35 (0.19)	[3.98, 4.72]
			Unfair	1.81 (0.09)	[1.62, 2.00]
		Dictator	Fair	4.63 (0.16)	[4.31, 4.96]
			Unfair	2.05 (0.12)	[1.81, 2.29]
	Individualistic	Proposer	Fair	4.02 (0.19)	[3.63, 4.40]
			Unfair	1.71 (0.10)	[1.52, 1.91]
		Dictator	Fair	4.90 (0.17)	[4.57, 5.23]
			Unfair	2.08 (0.13)	[1.83, 2.34]
Warmth	Prosocial	Proposer	Fair	3.81 (0.16)	[3.50, 4.11]
			Unfair	1.87 (0.08)	[1.71, 2.03]
		Dictator	Fair	4.49 (0.17)	[4.15, 4.82]
			Unfair	2.01 (0.12)	[1.78, 2.24]
	Individualistic	Proposer	Fair	3.58 (0.16)	[3.26, 3.90]
			Unfair	1.70 (0.08)	[1.53, 1.86]
		Dictator	Fair	4.64 (0.17)	[4.30, 4.98]
			Unfair	2.18 (0.12)	[1.94, 2.42]
Competence	Prosocial	Proposer	Fair	4.06 (0.13)	[3.80, 4.33]
			Unfair	3.51 (0.16)	[3.19, 3.83]
		Dictator	Fair	4.32 (0.13)	[4.07, 4.58]
			Unfair	4.00 (0.18)	[3.65, 4.35]
	Individualistic	Proposer	Fair	3.94 (0.14)	[3.66, 4.21]
			Unfair	3.55 (0.17)	[3.22, 3.89]
		Dictator	Fair	4.31 (0.13)	[4.04, 4.57]
			Unfair	4.11 (0.18)	[3.74, 4.47]

Note: standard errors, SE; CI, Confidence Interval; SVO, social value orientations.

**Table 2 behavsci-15-01537-t002:** The summary of mixed ANOVA results on social cognitions of “fair” vs. “unfair” proposers/dictators.

Dependent Variable	Effect	*F*	*p*	*η_p_* ^2^	CI_Low	CI_High
fairness	SVO	0.05	1	0.00	0.00	1
fairness	fairness reputation	686.15	<0.001	0.85	0.81	1
fairness	SVO × fairness reputation	0.00	1	0.00	0.00	1
fairness	Role	54.79	<0.001	0.31	0.21	1
fairness	SVO × Role	9.05	0.018	0.07	0.01	1
fairness	fairness reputation × Role	5.47	0.063	0.04	0.00	1
fairness	SVO × fairness reputation × Role	6.02	0.096	0.05	0.00	1
trustworthiness	SVO	0.01	1	0.00	0.00	1
trustworthiness	fairness reputation	393.18	<0.001	0.77	0.71	1
trustworthiness	SVO × fairness reputation	0.05	1	0.00	0.00	1
trustworthiness	Role	1.51	0.002	0.08	0.02	1
trustworthiness	SVO × Role	0.31	1	0.00	0.00	1
trustworthiness	fairness reputation × Role	14.29	<0.001	0.11	0.03	1
trustworthiness	SVO × fairness reputation × Role	0.22	1	0.00	0.00	1
altruism	SVO	0.58	1	0.00	0.00	1
altruism	fairness reputation	146.62	<0.001	0.55	0.45	1
altruism	SVO × fairness reputation	0.01	1	0.00	0.00	1
altruism	Role	15.62	<0.001	0.12	0.04	1
altruism	SVO × Role	0.64	1	0.01	0.00	1
altruism	fairness reputation × Role	31.22	<0.001	0.21	0.11	1
altruism	SVO × fairness reputation × Role	0.01	1	0.00	0.00	1
cooperation	SVO	0.06	1	0.00	0.00	1
cooperation	fairness reputation	445.67	<0.001	0.79	0.74	1
cooperation	SVO × fairness reputation	0.00	1	0.00	0.00	1
cooperation	Role	31.33	<0.001	0.21	0.11	1
cooperation	SVO × Role	5.29	0.095	0.04	0.00	1
cooperation	fairness reputation × Role	3.41	0.134	0.03	0.00	1
cooperation	SVO × fairness reputation × Role	2.34	0.645	0.02	0.00	1
warmth	SVO	0.02	0.885	0.00	0.00	0.00
warmth	fairness reputation	721.36	<0.001	0.79	0.67	0.63
warmth	SVO × fairness reputation	0.83	0.837	0.00	0.00	0.00
warmth	Role	51.99	<0.001	0.34	0.13	0.08
warmth	SVO × Role	4.78	0.029	0.04	0.01	0.00
warmth	fairness reputation × Role	11.71	<0.001	0.15	0.03	0.01
warmth	SVO × fairness reputation × Role	0.02	1	0.00	0.00	0.00
competence	SVO	0.00	1	0.00	0.00	1
competence	fairness reputation	9.75	0.002	0.08	0.02	1
competence	SVO × fairness reputation	0.41	1	0.00	0.00	1
competence	Role	24.60	<0.001	0.17	0.08	1
competence	SVO × Role	0.27	1	0.00	0.00	1
competence	fairness reputation × Role	2.94	0.134	0.02	0.00	1
competence	SVO × fairness reputation × Role	0.03	1	0.00	0.00	1

Note: ANOVA, Analysis of Variance; CI, Confidence Interval; SVO, social value orientations.

**Table 3 behavsci-15-01537-t003:** The means and standard errors (SE) on social cognitions of “fair” vs. “unfair” responders.

Social Cognition	SVO	Fairness Reputation	Mean (SE)	CI
Fairness	Prosocial	Fair	4.83 (0.20)	[4.42, 5.23]
		Unfair	2.57 (0.17)	[2.24, 2.90]
	Individualistic	Fair	5.00 (0.21)	[4.58, 5.42]
		Unfair	2.73 (0.17)	[2.39, 3.07]
Trustworthiness	Prosocial	Fair	4.29 (0.18)	[3.93, 4.64]
		Unfair	3.87 (0.21)	[3.46, 4.28]
	Individualistic	Fair	4.53 (0.18)	[4.16, 4.89]
		Unfair	3.92 (0.21)	[3.49, 4.34]
Altruism	Prosocial	Fair	2.56 (0.18)	[2.20, 2.91]
		Unfair	5.35 (0.21)	[4.93, 5.77]
	Individualistic	Fair	3.00 (0.19)	[2.63, 3.37]
		Unfair	5.39 (0.22)	[4.96, 5.82]
Cooperation	Prosocial	Fair	3.60 (0.20)	[3.22, 3.99]
		Unfair	5.13 (0.20)	[4.73, 5.53]
	Individualistic	Fair	3.97 (0.20)	[3.57, 4.37]
		Unfair	5.05 (0.21)	[4.64, 5.47]
Warmth	Prosocial	Fair	3.62 (0.17)	[3.28, 3.96]
		Unfair	5.12 (0.18)	[4.77, 5.47]
	Individualistic	Fair	3.97 (0.18)	[3.61, 4.32]
		Unfair	5.35 (0.18)	[4.98, 5.71]
Competence	Prosocial	Fair	4.35 (0.16)	[4.04, 4.67]
		Unfair	3.15 (0.16)	[2.83, 3.46]
	Individualistic	Fair	4.25 (0.16)	[3.92, 4.57]
		Unfair	3.21 (0.16)	[2.89, 3.54]

**Table 4 behavsci-15-01537-t004:** The summary of mixed ANOVA results on social cognitions of “fair” vs. “unfair” responders.

Dependent Variable	Effect	*F*	*p*	*η_p_* ^2^	CI_Low	CI_High
fairness	SVO	0.76	1	0.01	0.00	1
fairness	fairness reputation	141.75	<0.001	0.53	0.44	1
fairness	SVO × fairness reputation	0.002	1	0.00	0.00	1
reliable	SVO	0.44	1	0.00	0.00	1
reliable	fairness reputation	8.29	0.005	0.06	0.01	1
reliable	SVO × fairness reputation	0.31	1	0.00	0.00	1
altruism	SVO	1.32	1	0.01	0.00	1
altruism	fairness reputation	194.77	<0.001	0.62	0.53	1
altruism	SVO × fairness reputation	1.18	1	0.01	0.00	1
cooperation	SVO	0.44	1	0.00	0.00	1
cooperation	fairness reputation	48.56	<0.001	0.29	0.18	1
cooperation	SVO × fairness reputation	1.38	1	0.01	0.00	1
warmth	SVO	1.82	1	0.01	0.00	1
warmth	fairness reputation	112.34	<0.001	0.48	0.38	1
warmth	SVO × fairness reputation	0.19	1	0.00	0.00	1
competence	SVO	0.02	1	0.00	0.00	1
competence	fairness reputation	5.57	<0.001	0.30	0.19	1
competence	SVO × fairness reputation	0.3	1	0.00	0.00	1

Note: ANOVA, Analysis of Variance; CI, Confidence Interval; SVO, social value orientations.

## Data Availability

The data and code that support the findings of this study are openly available in Open Science Framework (OSF) at https://doi.org/10.17605/OSF.IO/G96AQ.

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
