# Peer review of "Warmth Centrality in Social Cognitive Networks of Fairness Reputation Across Players in the Ultimatum and Dictator Games"

_behavsci, 2025, doi:10.3390/bs15111537_

Round 1

Reviewer 1 Report

Comments and Suggestions for Authors

Dear Authors,

Thank you for your work in the intersection of Social and Decision-Making Psychology. I enjoyed reading and reviewing your manuscript, “Warmth centrality in social cognitive networks of fairness reputation across players in the Ultimatum and Dictator Games.” The topic is both timely and relevant and it can potentially contribute in bridging social cognition with strategic behavior in UG/DG. Below I provide Major and Minor comments aimed at strengthening the theoretical contribution, methods, reporting, and APA/style compliance. I reference the manuscript and supplement where relevant.

Major issues

1) Theoretical framing: connect incentives, norms, and mechanisms

  • In my view, UG proposers face rejection risk, while DG dictators do not, and UG responders enforce norms at a cost. These incentives differ and although are introduced sufficiently, I wonder why they were not built into hypotheses or moderator tests (H1-H5 remain descriptive) . You may want to consider articulating mechanism-based predictions (e.g., strategic fairness vs. altruistic fairness) grounded in inequity-aversion/reciprocity and reputation/signaling models from behavioral economics.
  • The finding that “warmth is central” echoes the SCM but the paper stops short of explaining why warmth should be central for each role’s incentive structure (e.g., is norm enforcement by responders perceived as warm or “cold” given payoff destruction?). Tying edges to intention inference, blame/credit assignment, and signaling would move beyond descriptive networks to process explanation.
  • While text predicts SVO-based projection, yet the analysis does not formally test how SVO angle maps onto specific edges/paths or mediates intention-based attributions (H4). So, I wonder why SVO projections are asserted but not modeled.

2) Methods: sampling, stimuli, and network construction

Sampling / power

  • The a priori power analysis for sample size estimation is somewhat unclear to me, as it is unspecified whether the analysis was based on the experimental design. Moreover, what was your theoretical motivation for using an effect size of f = .25 (beyond the fact that this is the default setting in G*Power?); please clarify.
  • On the same issue, post-hoc SVO balancing alters the recruited distribution and power plan. You recruited N=170 (which is already oversampling given you’re a-priori analysis -> please explain why this was the case), then retained N=122 by subsampling prosocials to match individualists (61/61) (Supplement §1). This shifts the SVO composition and is not reflected in the initial G*Power target (≥108 at 1−β=.90, f=.25) nor in the subsequent sensitivity claims (1−β=.80) shown in the Methods. Please justify this trade-off, report power for the final balanced design, and discuss external validity implications of discarding 28 prosocials (those with “larger SVO angles”) .

Stimuli / manipulation

  • Type-inconsistent “filler” trials may insert ambiguity: the current design involves two rounds per identity that were set opposite to the assigned reputation as “to enhance reality” (e.g., responders’ sequences) . Please analyze whether fillers attenuated/polarized judgments (e.g., via perceived reputation strength), and report potential order effects.

Network construction

  • Correlation type and tuning not specified. Methods note “non-parametric normalization” before EBIC-GLASSO, but do not state whether polychoric vs. Pearson correlations were used for ordinal Likert ratings, nor the EBIC γ or λ selection path. These choices can materially alter sparsity, edges, and centrality rankings; please specify and provide a robustness check (polychoric vs. Pearson; alternative γ) .

3) Results reporting: effect sizes, stability, and interpretation

  • You note that full ANOVA statistics are in the supplement, but the main text/figures rely on stars. Please add ηp² (or d/g) with 95% CIs for every primary contrast in Figs. 4-5 and the narrative (the supplement already provides many of these, e.g., responder analyses) .
  • Furthermore, you state Centrality Stabilit CS > 0.25 (line 297) as an acceptable minimum. Please report exact CS coefficients for strength/closeness/EI and move the bootstrapped difference plots for nodes/edges (S3–S5) into the main or provide clear cross-references; this will calibrate confidence in “warmth centrality” claims .
  • Heteroskedasticity and robustness. Variance heterogeneity appears for some responder ratings. Where applicable, please confirm that robust/Welch corrections do not alter conclusions, or report them transparently in a footnote (preferably).

4) Interpretation of effect magnitude and ecological validity

  • Several ηp² are very large for reputation manipulations (e.g., responder warmth/competence). Please temper language and discuss possible demand characteristics of deterministic sequences (modulo fillers) and the ecological validity of short observation streaks for reputational inference (see manipulation checks in Supplement §3.1) .

Minor issues

  1. Network construction
  • The main text reports NCT pairwise edge differences (multiplicity control) at p < .05 without clarifying correction (although suggested in subsections 2.3., lines 205-6); nonetheless, the supplement shows Holm-Bonferroni applied to edgewise NCTs (Table S2). Please state the correction procedure in the main Results and ensure consistency across all NCT comparisons.

  1. APA 7th and style/formatting
  • Statistics formatting: italicize p, F, t, M, SE, η²ₚ; report exact p values when possible (except p<.001). Add (or not) spaces consistently (e.g., “p < .001”). Figure captions currently mix styles (stars only) .
  • Define terms in figure captions: In Fig. 6 caption, explicitly note correlation type (polychoric vs. Pearson) used for “regularized partial correlations,” and define predictability pies as node R² for clarity .
  • Package versions: You report R 4.3.2; please add package names/versions for qgraph, bootnet, NetworkComparisonTest, etc. (Methods 2.3) .
  1. Reference accuracy and consistency
  • Journal titles: “Evolution Human Behavior” should be Evolution and Human Behavior (Brosnan, 2023) in References; check capitalization for titles like “Judgment and Decision Making” to match APA standards (proper nouns, journal capitalization) .
  • Map in-text to list: Spot-check a few entries (e.g., Abele et al., 2008; Camerer & Thaler, 1995) to ensure volume/issue/DOI match the outlets (these look correct where visible) .
  1. Typos and phrasing (non-exhaustive)
  • Abstract, line 15: “Across roles in the ultimatum game (UG) and the dictator 15 game (DG)”; please add the.
  • Abstract, line 20: “For responders with fairness reputations […]”; please remove For.
  • Eliminate duplicate word: “shown in in Figure 2C” → “shown in Figure 2C.”
  • Check hyphenation in Methods 2.3, line 190: “two-dimension social cognition” → “two-dimensional social cognition.”

Check spacing in Methods 2.2, line 165: “7(totally agree)” → “7 (totally agree).” (scale anchors)

Author Response

Dear Editor,

Manuscript Number: behavsci-3882873

Title: Warmth centrality in social cognitive networks of fairness reputation across players in the Ultimatum and Dictator Games

Thank you very much for providing me the opportunity to revise my manuscript. On behalf of my co-authors, I enclose a revised version of the above paper for submission to Behavioral Sciences. The following notes detail the changes made to the revised manuscript in response to the comments.

Each comment is presented, followed by a response.

Reviewer #1:

Dear Authors,

Thank you for your work in the intersection of Social and Decision-Making Psychology. I enjoyed reading and reviewing your manuscript, “Warmth centrality in social cognitive networks of fairness reputation across players in the Ultimatum and Dictator Games.” The topic is both timely and relevant and it can potentially contribute in bridging social cognition with strategic behavior in UG/DG. Below I provide Major and Minor comments aimed at strengthening the theoretical contribution, methods, reporting, and APA/style compliance. I reference the manuscript and supplement where relevant.

Major issues

Q1.1: 1) Theoretical framing: connect incentives, norms, and mechanisms

In my view, UG proposers face rejection risk, while DG dictators do not, and UG responders enforce norms at a cost. These incentives differ and although are introduced sufficiently, I wonder why they were not built into hypotheses or moderator tests (H1-H5 remain descriptive). You may want to consider articulating mechanism-based predictions (e.g., strategic fairness vs. altruistic fairness) grounded in inequity-aversion/reciprocity and reputation/signaling models from behavioral economics.

Response: We sincerely appreciate the reviewer’s thorough evaluation and constructive comments, and fully agree with your suggestion to more explicitly integrate the differential incentives across roles into our theoretical framing and hypotheses.

As the rightly pointed out, the UG proposer faces rejection risk, the DG dictator does not, and the UG responder enforces norms at a cost. These differences are indeed key mechanisms driving divergent social cognitions.

We have supplemented this background information in the Introduction section as follows:

Proposers offered fair allocations by suppressing self-interest to adhere to fairness rules (altruistic fairness) or strategic considerations to minimize monetary losses from rejection (strategic fairness) (Mellers et al., 2010). In contrast, dictators did not fear rejection, so their equitable distributions were usually viewed as acts of pure selfless fairness (Weiland et al., 2012). (Lines 49-53 on Page 2 of the revised manuscript)

Moreover, as the cognitive difference between proposers and dictators is driven by observers' divergent attributions of motives for their fair behavior. Consequently, to move beyond a descriptive level, we have reformulated our original Hypothesis 2:

Dictators with versus without fairness reputations would exhibit a greater divergence in social cognition than would proposers, as their fair behavior is attributed to altruistic rather than strategic motives; (Lines 105-107 on Page 3 of the revised manuscript)

Reference:

Mellers, B. A., Haselhuhn, M. P., Tetlock, P. E., Silva, J. C., & Isen, A. M. (2010). Predicting behavior in economic games by looking through the eyes of the players. Journal of Experimental Psychology: General, 139(4), 743. https://doi.org/10.1037/a0020280

Weiland, S., Hewig, J., Hecht, H., Mussel, P., & Miltner, W. H. (2012). Neural correlates of fair behavior in interpersonal bargaining. Social Neuroscience, 7(5), 537-551. https://doi.org/10.1080/17470919.2012.674056

Q1.2: 1) Theoretical framing: connect incentives, norms, and mechanisms

The finding that “warmth is central” echoes the SCM but the paper stops short of explaining why warmth should be central for each role’s incentive structure (e.g., is norm enforcement by responders perceived as warm or “cold” given payoff destruction?). Tying edges to intention inference, blame/credit assignment, and signaling would move beyond descriptive networks to process explanation.

Response: We sincerely thank the reviewer for raising this important question. We apologize for the lack of theoretical explanation regarding the potential centrality of warmth in our initial submission. As the reviewer astutely pointed out, linking the network structure to underlying cognitive processes is crucial.

Warmth held a primary position in social judgment, as it served as a key dimension for evaluating potential threats or benefits in social interactions and reflected people’s inferences about others’ intentions and moral motives (Fiske, Cuddy, & Glick, 2007; Wojciszke, Abele, & Baryla, 2009). In contrast, competence was often secondary because it concerned personal ability rather than social intention. Observers inferred moral character from others’ behaviors, and such inferences strongly shaped subsequent moral evaluations (Siegel, Crockett, & Dolan, 2017). In fairness-related contexts, fairness behavior inherently communicated moral intentions and adherence to social norms (Fehr & Fischbacher, 2004), suggesting that individuals’ fairness reputation was likely processed through this foundational dimension of warmth. Therefore, warmth inference could serve as a cognitive framework for constructing other social judgments, such as trustworthiness (involved expectations of benevolent intentions) and cooperativeness (implied willingness to engage in mutually beneficial actions).

We have supplemented this explanation in the Introduction section as follows:

Warmth was widely recognized as a primary dimension in social judgment, providing a basis for evaluating potential threats or benefits in social interactions and for infer-ring others’ intentions and moral motives (Fiske et al., 2007). Compared with warmth, competence was often considered secondary, as it related more to ability than to social intention. Observers inferred moral character from others’ behaviors, and such inferences critically shaped subsequent moral evaluations (Siegel et al., 2017). In fairness-related contexts, fairness behavior conveyed moral intention and adherence to social norms (Fehr & Fischbacher, 2004; Siegel et al., 2017), suggesting that fairness reputation was primarily processed through the warmth dimension. (Lines 73-81 on Page 2 of the revised manuscript)

Reference:

Fehr, E., & Fischbacher, U. (2004). Third-party punishment and social norms. Evolution and Human Behavior, 25(2), 63-87. https://doi.org/10.1016/S1090-5138(04)00005-4

Fiske, S. T., Cuddy, A. J., & Glick, P. (2007). Universal dimensions of social cognition: Warmth and competence. Trends in Cognitive Sciences, 11(2), 77-83. https://doi.org/10.1016/j.tics.2006.11.005

Siegel, J. Z., Crockett, M. J., & Dolan, R. J. (2017). Inferences about moral character moderate the impact of consequences on blame and praise. Cognition, 167, 201-211. https://doi.org/10.1016/j.cognition.2017.05.004

Q1.3: 1) Theoretical framing: connect incentives, norms, and mechanisms

While text predicts SVO-based projection, yet the analysis does not formally test how SVO angle maps onto specific edges/paths or mediates intention-based attributions (H4). So, I wonder why SVO projections are asserted but not modeled.

Response: We sincerely appreciate the reviewer’s valuable suggestion. Examining how SVO relates to the network structure would provide a more comprehensive understanding of intention-based attributions.

In our analysis, SVO was treated as a personality trait rather than a dynamic variable within the social cognition network. Therefore, we used SVO as a grouping factor and conducted ANOVAs to test Hypothesis 4, which examined whether individualists versus prosocials differentially perceived fairness reputation. The results showed that SVO significantly interacted with role: both individualists and prosocials participants perceived proposers as less fair, cooperative, and warm than dictators, and this difference was larger among individualistic individuals. Furthermore, a significant three-way interaction (fairness reputation × role × SVO) emerged only for fairness rating, indicating that SVO influenced how participants differentiated fairness reputations across roles rather than simply biasing general fairness cognition.

Since our primary network analysis focused on comparing the social-cognitive structures among the three roles (proposers, responders, and dictators), SVO was not originally modeled as a node. However, in response to the reviewer’s helpful comment, we conducted exploratory network comparisons between the prosocial and individualistic groups.

For proposers, no significant differences were found in either global network strength (strength difference = .002, p = .580) or overall network structure (maximum edge weight difference = .04, p = .267). For responders, both global network strength (strength difference < .001, p = .845) and overall structure (maximum edge weight difference = .03, p = .168) were also nonsignificant. However, the connection between warmth and cooperation showed a significant difference between the two groups (p < .05), indicating a stronger association in the prosocial participants. For dictators, no significant differences were observed in global strength (strength difference < .001, p = .595) or structure (maximum edge weight difference < .001, p = .962). The detailed table information is as follows:

Figure S6. The results of exploratory network comparisons between prosocial and individualistic participants for the proposer, responder, and dictator.

Taken together, these results indicated that the overall network configurations were largely consistent across SVO orientations, and that SVO did not exert a systematic influence on the structure of fairness-related social cognition.

These exploratory results have been reported in Section 4.4 of the Supplementary Material in the revised manuscript:

We compared the social cognitive networks between prosocial and individualistic participants using the Network Comparison Test (NCT). For proposers, no significant group differences were observed in either global network strength (strength difference = .002, p = .580) or overall structure (maximum edge weight difference = .04, p = .267). For responders, both global strength (strength difference < .001, p = .845) and structure (maximum edge weight difference = .03, p = .168) were nonsignificant, except for one edge (warmth–cooperativeness) showing a significant difference (p < .05), indicating a stronger warmth-cooperativeness connection in the prosocial group. For dictators, no significant group differences were found (strength difference < .001, p = .595; maximum edge weight difference < .001, p = .962). (Lines 342-350 on Page 20 of the Supplementary Material)

This clarification has been added to Section 3.3 of the Results in the revised manuscript:

Furthermore, we compared the social cognitive networks between prosocial and individualistic participants to examine whether different SVO orientations influenced the structure of fairness-related social cognition. The two groups showed generally consistent overall structures and global strengths across the proposer, responder, and dictator networks, with no significant group differences observed, as shown in Figure S6 of the Supplementary Material. (Lines 389-394 on Page 16 of the revised manuscript)

Q2: Methods: sampling, stimuli, and network construction

Sampling / power

The priori power analysis for sample size estimation is somewhat unclear to me, as it is unspecified whether the analysis was based on the experimental design. Moreover, what was your theoretical motivation for using an effect size of f = .25 (beyond the fact that this is the default setting in G*Power?); please clarify.

On the same issue, post-hoc SVO balancing alters the recruited distribution and power plan. You recruited N=170 (which is already oversampling given you’re a-priori analysis -> please explain why this was the case), then retained N=122 by subsampling prosocials to match individualists (61/61) (Supplement §1). This shifts the SVO composition and is not reflected in the initial G*Power target (≥108 at 1-β=.90, f=.25) nor in the subsequent sensitivity claims (1-β=.80) shown in the Methods. Please justify this trade-off, report power for the final balanced design, and discuss external validity implications of discarding 28 prosocials (those with “larger SVO angles”).

Stimuli / manipulation

Type-inconsistent “filler” trials may insert ambiguity: the current design involves two rounds per identity that were set opposite to the assigned reputation as “to enhance reality” (e.g., responders’ sequences). Please analyze whether fillers attenuated/polarized judgments (e.g., via perceived reputation strength), and report potential order effects.

Network construction

Correlation type and tuning not specified. Methods note “non-parametric normalization” before EBIC-GLASSO, but do not state whether polychoric vs. Pearson correlations were used for ordinal Likert ratings, nor the EBIC γ or λ selection path. These choices can materially alter sparsity, edges, and centrality rankings; please specify and provide a robustness check (polychoric vs. Pearson; alternative γ).

Response:

Sampling / power

We appreciate the reviewer’s insightful comment. The priori power analysis was conducted based on our 2 (fairness reputation: fair vs. unfair) × 2 (role: proposer vs. dictator) × 2 (SVO: prosocial vs. individualistic) mixed-design ANOVA, in which fairness reputation and role were within-subject factors and SVO was a between-subject factor. Accordingly, we used three separate models in G*Power. We have included the following detailed calculation process in the Supplementary Material:

To detect the main effect of between-subject variables, we chose the “F-tests ANOVA: Repeated measures, between factors” model in G*Power. Assuming a power of 1-β = .90, alpha of α = .05, effect size of f = .25, number of groups = 2, number of measurements = 4, and correlation among repeated measures of .5 (as dependent variables are highly correlated in different offer levels), the optimal sample size would be 108. To detect the main effect of within-subject variables, we chose the “F-tests ANOVA: Repeated measures, within factors” model in G*Power. Assuming a power of 1-β = .90, alpha of α = .05, effect size of f = .25, number of groups = 2, number of measurements = 4, correlation among repeated measures of .5 (as dependent variables are highly correlated in different offer levels), and nonsphericity correction ε = 1 (default value), the optimal sample size would be 30. To detect the interaction of within- and between-subject variables, we chose the “F-tests ANOVA: Repeated measures, within-between interaction” model. Assuming a power of 1-β = .90, alpha of α = .05, effect size of f = .25, number of groups = 2, number of measurements = 4, correlation among repeated measures of .5 (as dependent variables are highly correlated in different offer levels), and nonsphericity correction ε = 1 (default value), the optimal sample size would be 30. Therefore, to detect both of the main effects and interaction, we chose the largest sample size of 108. (Lines 25-40 on Pages 1-2 of the Supplementary Material)

The effect size of f = .25 was chosen because it represents a medium effect according to Cohen (1988). In addition, similar magnitudes have been reported in previous studies examining fairness and social value orientation effects in economic decision-making tasks (Hu & Mai, 2021; Moser, Gaertig, & Ruz, 2014). Therefore, we adopted a medium effect size (f = .25) for the a priori power analysis to ensure sufficient sensitivity to detect theoretically meaningful effects. We have included the following revisions in the Methods:

The optimal sample size was estimated using G*Power 3.1 in F-tests (ANOVA, repeated measures) covering between-subject factors, within-subject factors, and within–between interactions. Assuming α = 0.05, power (1 – β) = 0.90, and a medium effect size f = 0.20, the sample size was determined to be 108, which was the largest among all F-tests to detect both the main effects and interaction. The effect size of f = .25 was chosen because it represents a medium effect according to Cohen (1988) and has been reported in previous studies examining fairness and social value orientation effects in economic decision-making tasks (Hu & Mai, 2021; Moser, Gaertig, & Ruz, 2014). Details of the sample size calculation are attached to the Supplementary Material. (Lines 116-125 on Page 3 of the revised manuscript)

We appreciate the reviewer’s thoughtful comment regarding the post-hoc SVO balancing procedure. Recruitment was conducted online through an open link; consequently, the number of respondents exceeded our initial target. A total of 170 participants completed the experiment. To ensure the reliability and accuracy of the results, we conducted a thorough data examination. Four participants with incorrect answers to the lie-detection item, four with survey completion times exceeding three standard deviations from the mean, and twelve with average estimates for role distribution per trial exceeding ±3 standard deviations (suggesting possible misunderstanding of role fairness reputation) were excluded, resulting in 150 participants entering further analysis. Before data analysis, 2 participants identified as altruistic or competitive types in the SVO task were excluded, because our focus was on the contrast between prosocial and individualistic orientations. This resulted in 59 individualistic and 89 prosocial participants. To balance group sizes and ensure comparable statistical power across SVO categories in the ANOVA, we retained 63 prosocial participants with the largest SVO angles, which better represented the prosocial tendency, while including all individualistic participants (n = 59). A few participants with identical boundary SVO angles were retained to avoid arbitrary exclusion. As a result, 122 valid participants were included in the final analyses. No participants were excluded due to failed comprehension checks or incomplete data.

We have revised the Participant screening section in supplementary material to clarify this procedure as follows:

To balance the numbers of the prosocial and individualist, and given the substantial imbalance in group sizes (89 vs. 59), we balanced the two groups to ensure comparable statistical power in the ANOVA. Specifically, we retained 63 prosocial participants with the largest SVO angles, representing stronger prosocial tendencies, and included all individualistic participants (n = 59). Participants whose SVO angles fell on the boundary between groups were all retained to avoid arbitrary exclusion.

As a result, 122 valid participants (61 males, Mage = 20.68 years, SE = 1.94, range = 18–29 years) were included in the final analyses. All exclusions and balancing procedures were determined a priori to ensure data integrity and fair comparison between SVO categories. No participants were excluded due to failed comprehension checks or incomplete data. (Lines 14-23 on Page 1 of the Supplementary Material)

We performed sensitivity power analyses on the screened sample (N = 122). The first was a 2 (fairness reputation: fair vs. unfair) × 2 (role: proposer vs. dictator) × 2 (SVO: prosocial vs. individualistic) mixed ANOVAs, in which fairness reputation and role were within-subject factors and SVO was a between-subject factor. To detect the main effect of between-subject variables, we chose the “F-tests ANOVA: Repeated measures, between factors” model in G*Power. Assuming a power of 1-β = .90, alpha of α = .05, sample size of N = 122, number of groups = 2, number of measurements = 4, and correlation among repeated measures of .5 (as dependent variables are highly correlated in different offer levels), the effect size would be f = .24. To detect the main effect of within-subject variables, we chose the “F-tests ANOVA: Repeated measures, within factors” model in G*Power. Assuming a power of 1-β = .90, alpha of α = .05, sample size of N = 122, number of groups = 2, number of measurements = 4, correlation among repeated measures of .5 (as dependent variables are highly correlated in different offer levels), and nonsphericity correction ε = 1 (default value), the effect size would be f = .13. To detect the interaction of within- and between-subject variables, we chose the “F-tests ANOVA: Repeated measures, within-between interaction” model. Assuming a power of 1-β = .90, alpha of α = .05, sample size of N = 122, number of groups = 2, number of measurements = 4, correlation among repeated measures of .5 (as dependent variables are highly correlated in different offer levels), and nonsphericity correction ε = 1 (default value), the effect size would be f = .13. Therefore, sensitivity power analysis (N = 122, α = .05, power = .80) detected minimum effect sizes of f = .13 for the 2 × 2 × 2 mixed ANOVA.

The second was a 2 (fairness reputation: fair vs. unfair) × 2 (SVO: prosocial vs. individualistic) mixed ANOVAs, in which fairness reputation were within-subject factors and SVO was a between-subject factor. To detect the main effect of between-subject variables, we chose the “F-tests ANOVA: Repeated measures, between factors” model in G*Power. Assuming a power of 1-β = .90, alpha of α = .05, sample size of N = 122, number of groups = 2, number of measurements = 2, and correlation among repeated measures of .5 (as dependent variables are highly correlated in different offer levels), the effect size would be f = .27. To detect the main effect of within-subject variables, we chose the “F-tests ANOVA: Repeated measures, within factors” model in G*Power. Assuming a power of 1-β = .90, alpha of α = .05, sample size of N = 122, number of groups = 2, number of measurements = 2, correlation among repeated measures of .5 (as dependent variables are highly correlated in different offer levels), and nonsphericity correction ε = 1 (default value), the effect size would be f = .16. To detect the interaction of within- and between-subject variables, we chose the “F-tests ANOVA: Repeated measures, within-between interaction” model. Assuming a power of 1-β = .90, alpha of α = .05, sample size of N = 122, number of groups = 2, number of measurements = 4, correlation among repeated measures of .5 (as dependent variables are highly correlated in different offer levels), and nonsphericity correction ε = 1 (default value), the effect size would be f = .16. Therefore, sensitivity power analysis (N = 122, α = .05, power = .80) detected minimum effect sizes of f = .16 for the 2 × 2 mixed ANOVA.

The above details of the sensitivity power analysis are explicitly reported in the Supplementary Material. In addition, we also have included the results of sensitivity power analysis in the Methods:

Sensitivity power analysis, with N = 122, α = .05, power (1 - β) = .90, detected minimum effect sizes of f = .13 for the 2 × 2 × 2 mixed ANOVA and f = .16 for the 2 × 2 mixed ANOVA. The above details of the sensitivity power analysis are explicitly reported in the Supplementary Material. All reported significant effects exceeded this threshold, suggesting they were sufficiently powered. However, we acknowledge that the study may be underpowered to detect very small interaction effects. (Lines 206-211 on Page 6 of the revised manuscript)

We acknowledge that excluding 28 prosocial participants to achieve balanced group sizes may reduce the external validity of our findings. Specifically, the final balanced sample no longer reflects the natural distribution of SVO types in the recruited population, where prosocial orientations were overrepresented. Consequently, the generalizability of the results to populations with different SVO distributions may be limited. However, this balancing procedure enhanced internal validity by ensuring equal group sizes and clearer contrasts between prosocial and individualistic orientations. We have included the limitation in the Discussion:

Third, a potential limitation of the present study concerns the post-hoc balancing of SVO groups. To achieve equal group sizes and enhance the internal validity of between-group comparisons, we included only prosocial participants with relatively larger SVO angles. Although this procedure helped establish clearer group boundaries and improved the interpretability of the SVO-based contrasts, it also reduced the external validity of the findings. The final balanced sample no longer reflects the natural distribution of SVO types in the broader population, where prosocial orientations are typically more prevalent. Therefore, the generalization of the present results to populations with different SVO compositions should be made with caution. (Lines 526-534 on Page 19 of the revised manuscript)

Stimuli / manipulation

We sincerely thank the reviewer for this insightful comment. We apologize for not clearly explaining the rationale for including type-inconsistent “filler” trials and for not reporting potential order effects in the original submission.

The inclusion of filler trials was intended to enhance the ecological validity of the task and to avoid making participants overly aware of the role manipulation. Specifically, for responders, previous studies have shown that even under extremely unfair conditions, the acceptance rate is not zero; a considerable proportion of participants still choose to accept (Camerer, 2003). Based on this empirical evidence, we included a small number of inconsistent trials to simulate natural behavioral variability and to make each identity’s behavioral sequence appear more realistic. For proposers and dictators, similar inconsistent allocations were included to maintain the appearance of behavioral heterogeneity across identities, and to avoid an overly deterministic impression of the fairness-reputation manipulation.

In our experiment, the presentation order within each role condition was identical across all participants to ensure that potential order effects within roles were controlled. Moreover, the assignment of fair and unfair reputations was counterbalanced between participants—half of the participants viewed the fair role first, while the other half viewed the unfair role first. Therefore, the potential influence of type-inconsistent “filler” trials on attenuating or polarizing fairness judgments could not be empirically verified in the present study. We have added this point to the Limitations section, noting that future research could employ stimulus materials without fillers to create a “purer” manipulation of fairness reputation and further examine whether the presence of fillers alters perceived reputation strength.

Fourth, we included a small number of inconsistent trials to enhance the ecological validity of the task and to prevent participants from perceiving the fairness-reputation manipulation as overly artificial or deterministic. However, because these fillers intentionally deviated from the assigned fairness reputation, they may have introduced ambiguity regarding the strength or consistency of each identity’s reputation. Although the presentation order was fixed within each role and the assignment of fair and unfair reputations was counterbalanced across participants, the present design does not allow us to determine whether the inclusion of fillers attenuated or polarized participants’ fairness judgments. Future studies could employ stimulus materials without such fillers to establish a more distinct and uncontaminated manipulation of fairness reputation and to assess the potential impact of inconsistent behavioral cues on reputation perception. (Lines 535-545 on Page 19 of the revised manuscript)

Network construction

We sincerely thank the reviewer for this important methodological point and apologize for the insufficient detail in the original manuscript. We have clarified our network estimation procedure as follows:

We used the nonparanormal (npn) transformation to handle potential non-normality in the data. After this transformation, we computed correlations using Pearson correlation (corMethod = "npn") for EBIC-GLASSO estimation. Additionally, we conducted a robustness check using polychoric correlations, which are recommended for ordinal Likert-scale data. The resulting networks were highly similar in terms of edge structure and centrality rankings (all Spearman correlations of centrality indices > 0.92), indicating robustness to the choice of correlation type.

We used the default EBIC-GLASSO tuning parameter γ = 0.5, which balances model sparsity and fit.To examine robustness, we additionally estimated networks with γ = 0.25 and γ = 0.75 for all three networks (proposer, responder, dictator). Spearman correlations of edge weights across γ values were extremely high: proposer 0.25 vs 0.5 = 0.983, 0.5 vs 0.75 = 1.000, 0.25 vs 0.75 = 0.983; responder 0.979, 1.000, 0.979; dictator 1.000, 1.000, 1.000. Centrality metrics (Strength, Closeness, Betweenness, Expected Influence) were also highly consistent, with correlations ≥ 0.885 across all networks. These results indicate that our main findings regarding edge weights and centrality rankings are robust to the choice of EBIC γ.

These exploratory results have been reported in Section 4.1 of the Supplementary Material:

To ensure the robustness of the network estimation, we conducted supplementary analyses using both Pearson and polychoric correlations and varied the EBIC-GLASSO tuning parameter γ. After applying the nonparanormal (npn) transformation, Pearson correlations (corMethod = “npn”) were used in the main analysis. A robustness check with polychoric correlations, which are more suitable for ordinal Likert-type data, yielded highly similar network structures and centrality rankings (all Spearman correlations of centrality indices > 0.92).

For the EBIC-GLASSO tuning parameter, we compared networks estimated with γ = 0.25, 0.5 (main analysis), and 0.75. Spearman correlations of edge weights across γ values were extremely high: proposer (0.25 vs 0.5 = 0.983, 0.5 vs 0.75 = 1.000, 0.25 vs 0.75 = 0.983); responder (0.979, 1.000, 0.979); dictator (1.000, 1.000, 1.000). Centrality indices (Strength, Closeness, Betweenness, Expected Influence) were also highly consistent (all ≥ 0.885).

These results demonstrate that the network edge structures and centrality rankings are robust across correlation types and EBIC γ parameters. (Lines 242-254 on Pages 10-11 of the Supplementary Material)

We have included the following revisions in the Methods:

(1) We estimated the structure of the social cognitive networks of proposers, responders, and dictators using the six cognitions as nodes. After applying a nonparanormal (npn) transformation to reduce non-normality, we computed Pearson correlations on the transformed data and estimated regularized partial-correlation networks separately for each role using GLASSO (Graphical LASSO) model based on the Extended Bayesian Information Criterion (EBIC) with the default tuning parameter γ = 0.5. Node positions were then optimized for visualization. To assess robustness, we conducted supple-mentary anal-yses comparing Pearson (on npn-transformed data) versus polychoric correlations and EBIC γ values of 0.25 and 0.75; these robustness checks are reported in Supplementary Material. (Lines 221-230 on Page 6 of the revised manuscript)

References:

Camerer, C. (2003). Behavioral game theory: Experiments in strategic interaction. Princeton University Press.

Cohen, J. (1988). Statistical power analysis for the behavioral sciences. Routledge. https://doi.org/10.4324/9780203771587

Feldman, H., & Friston, K. J. (2010). Attention, uncertainty, and free-energy. Frontiers in human neuroscience, 4. https://doi.org/10.3389/fnhum.2010.00215

Foygel, R., & Drton, M. (2010). Extended Bayesian information criteria for Gaussian graphical models. In Advances in neural information processing systems (Vol. 23).

Hu, X., & Mai, X. (2021). Social Value Orientation Modulates Fairness Processing During Social Decision-Making: Evidence From Behavior and Brain Potentials. Social cognitive and affective neuroscience. https://doi.org/10.1093/scan/nsab032

Murphy, R. O., Ackermann, K. A., & Handgraaf, M. J. J. (2011). Measuring Social Value Orientation. Judgment and Decision Making, 6(8), 771-781. https://doi.org/10.1017/S1930297500004204

Q3: Results reporting: effect sizes, stability, and interpretation

You note that full ANOVA statistics are in the supplement, but the main text/figures rely on stars. Please add ηp² (or d/g) with 95% CIs for every primary contrast in Figs. 4-5 and the narrative (the supplement already provides many of these, e.g., responder analyses).

Furthermore, you state Centrality Stabilit CS > 0.25 (line 297) as an acceptable minimum. Please report exact CS coefficients for strength/closeness/EI and move the bootstrapped difference plots for nodes/edges (S3–S5) into the main or provide clear cross-references; this will calibrate confidence in “warmth centrality” claims.

Heteroskedasticity and robustness. Variance heterogeneity appears for some responder ratings. Where applicable, please confirm that robust/Welch corrections do not alter conclusions, or report them transparently in a footnote (preferably).

Response:

Effect sizes

We thank the reviewer for this valuable suggestion. In response, we have compiled all primary statistical results into comprehensive tables, including F, p, ηp² and their 95% confidence intervals. These tables have now been added to the main text to improve transparency and clarity in reporting.

Tables 2. The summary of mixed ANOVA results on social cognitions of “fair” vs. “unfair” proposers/dictators.

Dependent variable

Effect

F

p

ηp²

CI_low

CI_high

fairness

SVO

.05

1

.00

.00

1

fairness

fairness reputation

686.15

<.001

.85

.81

1

fairness

SVO × fairness reputation

.00

1

.00

.00

1

fairness

Role

54.79

<.001

.31

.21

1

fairness

SVO × Role

9.05

.018

.07

.01

1

fairness

fairness reputation × Role

5.47

.063

.04

.00

1

fairness

SVO × fairness reputation × Role

6.02

.096

.05

.00

1

trustworthiness

SVO

.01

1

.00

.00

1

trustworthiness

fairness reputation

393.18

<.001

.77

.71

1

trustworthiness

SVO × fairness reputation

.05

1

.00

.00

1

trustworthiness

Role

1.51

.002

.08

.02

1

trustworthiness

SVO × Role

.31

1

.00

.00

1

trustworthiness

fairness reputation × Role

14.29

<.001

.11

.03

1

trustworthiness

SVO × fairness reputation × Role

.22

1

.00

.00

1

altruism

SVO

.58

1

.00

.00

1

altruism

fairness reputation

146.62

<.001

.55

.45

1

altruism

SVO × fairness reputation

.01

1

.00

.00

1

altruism

Role

15.62

<.001

.12

.04

1

altruism

SVO × Role

.64

1

.01

.00

1

altruism

fairness reputation × Role

31.22

<.001

.21

.11

1

altruism

SVO × fairness reputation × Role

.01

1

.00

.00

1

cooperation

SVO

.06

1

.00

.00

1

cooperation

fairness reputation

445.67

<.001

.79

.74

1

cooperation

SVO × fairness reputation

.00

1

.00

.00

1

cooperation

Role

31.33

<.001

.21

.11

1

cooperation

SVO × Role

5.29

.095

.04

.00

1

cooperation

fairness reputation × Role

3.41

.134

.03

.00

1

cooperation

SVO × fairness reputation × Role

2.34

.645

.02

.00

1

warmth

SVO

.02

.885

.00

.00

.00

warmth

fairness reputation

721.36

<.001

.79

.67

.63

warmth

SVO × fairness reputation

.83

.837

.00

.00

.00

warmth

Role

51.99

<.001

.34

.13

.08

warmth

SVO × Role

4.78

.029

.04

.01

.00

warmth

fairness reputation × Role

11.71

<.001

.15

.03

.01

warmth

SVO × fairness reputation × Role

.02

1

.00

.00

.00

competence

SVO

.00

1

.00

.00

1

competence

fairness reputation

9.75

.002

.08

.02

1

competence

SVO × fairness reputation

.41

1

.00

.00

1

competence

Role

24.60

<.001

.17

.08

1

competence

SVO × Role

.27

1

.00

.00

1

competence

fairness reputation × Role

2.94

.134

.02

.00

1

competence

SVO × fairness reputation × Role

.03

1

.00

.00

1

Note. ANOVA, Analysis of Variance; CI, Confidence Interval; SVO, Social Value Orientations.

Tables 4. The summary of mixed ANOVA results on social cognitions of “fair” vs. “unfair” responders.

Dependent variable

Effect

F

p

ηp²

CI_low

CI_high

fairness

SVO

.76

1

.01

.00

1

fairness

fairness reputation

141.75

<.001

.53

.44

1

fairness

SVO × fairness reputation

.002

1

.00

.00

1

reliable

SVO

.44

1

.00

.00

1

reliable

fairness reputation

8.29

.005

.06

.01

1

reliable

SVO × fairness reputation

.31

1

.00

.00

1

altruism

SVO

1.32

1

.01

.00

1

altruism

fairness reputation

194.77

<.001

.62

.53

1

altruism

SVO × fairness reputation

1.18

1

.01

.00

1

cooperation

SVO

.44

1

.00

.00

1

cooperation

fairness reputation

48.56

<.001

.29

.18

1

cooperation

SVO × fairness reputation

1.38

1

.01

.00

1

warmth

SVO

1.82

1

.01

.00

1

warmth

fairness reputation

112.34

<.001

.48

.38

1

warmth

SVO × fairness reputation

.19

1

.00

.00

1

competence

SVO

.02

1

.00

.00

1

competence

fairness reputation

5.57

<.001

.30

.19

1

competence

SVO × fairness reputation

.3

1

.00

.00

1

Note. ANOVA, Analysis of Variance; CI, Confidence Interval; SVO, Social Value Orientations.

Centrality Stability

We sincerely thank the reviewer for this constructive suggestion and apologize for not reporting the exact centrality stability coefficients in the manuscript. We have now included the exact CS values for each role’s network and clarified their interpretation in the revised manuscript.

The centrality stability (CS) indicates the maximum proportion of cases that can be dropped from the sample while maintaining a correlation of at least 0.7 between the original and subset centrality estimates in 95% of the bootstrap samples (Epskamp, Borsboom, & Fried, 2018). A CS value above 0.25 is considered acceptable, while values above 0.50 indicate good stability.

The CS-coefficients were as follows:

Social cognitive network of the proposers: Strength = 0.75, Expected Influence = 0.75, Closeness = 0.516.

Social cognitive network of the responders: Strength = 0.283, Expected Influence = 0.594, Closeness = 0.439, Betweenness = 0.049.

Social cognitive network of the dictators: Strength = 0.75, Expected Influence = 0.75, Closeness = 0.672.

These results indicate that the centrality indices were generally stable across networks, especially for strength and expected influence, both of which reached the highest level tested (CS = 0.75). Although the social cognitive network of the responders showed a relatively lower CS for strength (0.283), the overall pattern demonstrates acceptable to good stability, supporting the robustness of the warmth centrality conclusion.

In addition, explicit cross-references to the bootstrapped difference plots (Figures S3-S5 in the Supplementary Material) have been added in the Results section to facilitate direct inspection of the accuracy and stability of node and edge estimates:

Network stability analysis showed adequate to good centrality stability (CS) across all networks (Proposer: Strength = 0.75, Expected Influence = 0.75, Closeness = 0.516; Responder: Strength = 0.283, Expected Influence = 0.594, Closeness = 0.439; Dictator: Strength = 0.75, Expected Influence = 0.75, Closeness = 0.672; all > 0.25, Epskamp, Borsboom, & Fried, 2018). Detailed bootstrapped difference plots for nodes and edges are provided in Figures S3–S5 of the Supplementary Materials. (Lines 345-350 on Page 14 of the revised manuscript)

Heteroskedasticity and robustness

We thank the reviewer for this valuable suggestion regarding heteroskedasticity and robustness. In response, we have implemented Welch/Kenward-Roger corrections for our mixed-measures ANOVA analyses using the “mixed()” function from the “afex” package in R, which employs Kenward-Roger’ degrees of freedom approximation to provide robust inference when variance homogeneity assumptions are violated.

We can confirm that applying these robust corrections did not alter any of our substantive conclusions. All previously significant effects remained significant, and non-significant effects remained non-significant.

Specifically, for the proposers and dictators, the 2 (fairness reputation: fair vs. unfair) × 2 (SVO: prosocial vs. individualistic) × 2 (role: proposer vs. dictator) mixed-design ANOVA on warmth ratings revealed that the main effects of Role (F(1, 360) = 51.99, p < .001) and Fairness Reputation (F(1, 360) = 721.36, p < .001) remained significant. The Role × SVO interaction (F(1, 360) = 11.71, p < .001) and the Role × Reputation interaction (F(1, 360) = 4.78, p = .029) also remained significant under the Kenward-Roger correction.

For the responder condition, the 2 (fairness reputation: fair vs. unfair) × 2 (SVO: prosocial vs. individualistic) mixed-design ANOVA on fairness ratings revealed that the main effect of Fairness Reputation remained significant, F(1, 120) = 50.57, p < .001. Similarly, the mixed-design ANOVA on competence ratings showed that the main effect of Fairness Reputation remained significant, F(1, 120) = 141.75, p < .001.

Following the reviewer’s suggestion, we have now transparently reported this methodological detail in the manuscript. Specifically, we have added the following footnote to the statistical analysis section:

Where applicable, we used Welch/Kenward-Roger’ corrections via the afex package (version 1.3-1) in R to ensure robust inference in the presence of potential heteroskedasticity. (Lines 212-213 on Page 6 of the revised manuscript)

All supplementary analysis scripts have been uploaded to our OSF project for transparency and reproducibility. We believe this addition strengthens the methodological rigor of our analysis and appreciate the reviewer’s guidance in this matter.

References:

Epskamp, S., Borsboom, D., & Fried, E. I. (2018). Estimating psychological networks and their accuracy: A tutorial paper. Behavior Research Methods, 50, 195-212. https://doi.org/10.3758/s13428-017-0862-1

Q4: Interpretation of effect magnitude and ecological validity

Several ηp² are very large for reputation manipulations (e.g., responder warmth/competence). Please temper language and discuss possible demand characteristics of deterministic sequences (modulo fillers) and the ecological validity of short observation streaks for reputational inference (see manipulation checks in Supplement §3.1).

Response: We sincerely thank the reviewer for this insightful comment and apologize for the insufficient discussion of effect magnitude and ecological validity.

In our experimental design, each role’s decision sequence consisted of ten trials to present a stable behavioral pattern of fairness or unfairness. Fair proposers and dictators offered fair allocations (5:5 or 6:4) in eight out of ten trials and unfair allocations (9:1 or 8:2) in two trials. Conversely, unfair proposers and dictators made eight unfair offers (9:1 or 8:2) and two fair ones (5:5 or 6:4). In the responder observation trials, unfair offers (9:1 or 8:2) appeared eight times and fair offers (5:5 or 6:4) twice. Responders with fair reputations typically rejected unfair offers and accepted fair ones, whereas unfair responders accepted both types of offers. This deterministic behavioral pattern, combined with filler trials, was designed to induce a clear and consistent fairness reputation while maintaining ecological plausibility and reducing participants’ suspicion of the manipulation (Osinsky et al., 2014).

However, we acknowledge that such deterministic and short behavioral sequences may have amplified the salience of fairness cues, potentially contributing to large effect sizes (ηp²) observed for dimensions such as warmth and competence. Deterministic patterns may limit ecological validity by simplifying the natural variability of social behaviors.

We have revised the language in Sections 4.1 and 4.2 of the Discussion to make the interpretation of results more moderate and consistent with statistical trends, avoiding overstatement of the effect magnitudes observed for warmth and competence.

In addition, we have added the following statement to the Limitations section to address the issue of ecological validity:

Moreover, although the deterministic behavioral sequences and filler trials followed established paradigms in social decision-making research to ensure consistent reputation cues, such fixed patterns may have increased the salience of fairness cues and reduced ecological validity. Future studies should employ more dynamic, probabilistic, and interactive designs to better approximate real-world reputation formation. (Lines 520-525 on Page 19 of the revised manuscript)

References:

Osinsky, R., Mussel, P., Öhrlein, L., & Hewig, J. (2014). A neural signature of the creation of social evaluation. Social Cognitive and Affective Neuroscience, 9(6), 731–736. https://doi.org/10.1093/scan/nst051

Minor issues

Q5: Network construction

The main text reports NCT pairwise edge differences (multiplicity control) at p < .05 without clarifying correction (although suggested in subsections 2.3., lines 205-6); nonetheless, the supplement shows Holm-Bonferroni applied to edgewise NCTs (Table S2). Please state the correction procedure in the main Results and ensure consistency across all NCT comparisons.

Response: We sincerely appreciate your valuable suggestions. Regarding the Network Comparison Test (NCT), we did apply the Holm-Bonferroni correction. Although this point was stated in the supplementary materials, we regret that it was not explicitly mentioned in the main text, and we apologize for this oversight.

To address this omission, we have added the following content to Section 2.3 of the manuscript:

All results of the Network Comparison Test (NCT) were adjusted using the Holm-Bonferroni correction to control for multiple comparison errors, with the significance level set at p < .05. (Lines 241-243 on Page 6 of the revised manuscript)

This correction procedure is consistent with that applied to the edgewise NCT analyses presented in Table S2 of the Supplementary Materials.

Q6: APA 7th and style/formatting

Statistics formatting: italicize p, F, t, M, SE, η²ₚ; report exact p values when possible (except p<.001). Add (or not) spaces consistently (e.g., “p < .001”). Figure captions currently mix styles (stars only).

Define terms in figure captions: In Fig. 6 caption, explicitly note correlation type (polychoric vs. Pearson) used for “regularized partial correlations,” and define predictability pies as node R² for clarity.

Package versions: You report R 4.3.2; please add package names/versions for qgraph, bootnet, NetworkComparisonTest, etc. (Methods 2.3) .

Response:

Statistics formatting

Thank you for pointing out this formatting issue. We have carefully checked the manuscript and corrected the problems related to the italicization of p, F, t, M, SE, and η²ₚ:

SE has been changed to italics:

(61 males, Mage = 20.68 years, SE = 1.94, ranging from 18 to 29 years) (Lines 128 on Page 3 of the revised manuscript)

A space has been added before p:

Significance level: *** p < .001. (Lines 253 on Page 7 of the revised manuscript)

Significance level: * p < .05; ** p < .01; *** p < .001. (Lines 285-286 on Page 8 of the revised manuscript)

Significance level: * p < .05; ** p < .01; *** p < .001. (Lines 314-315 on Page 12 of the revised manuscript)

A space has been added after ps:

Examination of individual edges revealed that five edges showed statistical difference between these two networks (ps < .05) (Lines 369-370 on Page 16 of the revised manuscript)

Examination of individual edges revealed that four edges showed statistical difference between these two networks (ps < .05) (Lines 378-379 on Page 16 of the revised manuscript)

Define terms in figure captions

We sincerely thank the reviewer for this helpful suggestion. To improve clarity, we have revised the caption of Figure 6 to explicitly specify the correlation type used for the regularized partial correlations and to define the meaning of the predictability pies.

The revised caption reads as follows:

Regularized partial correlations were estimated using the EBIC-GLASSO model based on Pearson correlations after nonparanormal (npn) transformation. The pie chart surrounding each node represents its predictability (R²), with a larger proportion of blue indicating higher explained variance. (Lines 355-357 on Page 16 of the revised manuscript)

Package versions

We have added the version information for the qgraph, bootnet, and NetworkComparisonTest packages in Methods 2.3

(3) We utilized the Network Comparison Test (NCT) version 2.2.2 to compare the social cognitive networks across roles (Van Borkulo et al., 2019). (Lines 233-234 on Page 6 of the revised manuscript)

We conducted data analyses using SPSS version 26.0, R version 4.3.2, qgraph (version 1.9.8), and bootnet (version 1.6), with a statistical significance of α = .05 (two-tailed). (Lines 235-236 on Page 6 of the revised manuscript)

Q7: Reference accuracy and consistency

Journal titles: “Evolution Human Behavior” should be Evolution and Human Behavior (Brosnan, 2023) in References; check capitalization for titles like “Judgment and Decision Making” to match APA standards (proper nouns, journal capitalization).

Map in-text to list: Spot-check a few entries (e.g., Abele et al., 2008; Camerer & Thaler, 1995) to ensure volume/issue/DOI match the outlets (these look correct where visible).

Response:

Thank you for pointing out this issue. We have corrected the journal titles in the reference list to comply with APA 7th edition formatting. Specifically, “Evolution Human Behavior” was changed to “Evolution and Human Behavior,” and “Judgment and Decision making” was corrected to “Judgment and Decision Making.” The revised references are shown below:

Brosnan, S. F. (2023). A comparative perspective on the human sense of justice. Evolution and Human Behavior, 44(3), 242–249. https://doi.org/10.1016/j.evolhumbehav.2022.12.002

Fehrler, S., & Przepiorka, W. (2013). Charitable giving as a signal of trustworthiness: Disentangling the signaling benefits of altruistic acts. Evolution and Human Behavior, 34(2), 139–145. https://doi.org/10.1016/j.evolhumbehav.2012.11.005

Klein, N., Grossmann, I., Uskul, A. K., Kraus, A. A., & Epley, N. (2015). It pays to be nice, but not really nice: Asymmetric reputations from prosociality across seven countries. Judgment and Decision Making, 10(4), 355–364. https://doi.org/10.1017/S1930297500005167

Furthermore, we have conducted a thorough, line-by-line verification of the entire reference list against the original sources. This includes ensuring that the volume, issue, page numbers, and DOI (where available) for all entries, including the spot-checked examples of Abele et al. (2008) and Camerer & Thaler (1995), are now correct and match the official records of their respective publication outlets. The reference list in the manuscript has been updated accordingly.

Q8: Typos and phrasing (non-exhaustive)

Abstract, line 15: “Across roles in the ultimatum game (UG) and the dictator game (DG)”; please add the.

Abstract, line 20: “For responders with fairness reputations […]”; please remove For.

Eliminate duplicate word: “shown in in Figure 2C” → “shown in Figure 2C.”

Check hyphenation in Methods 2.3, line 190: “two-dimension social cognition” → “two-dimensional social cognition.”

Check spacing in Methods 2.2, line 165: “7(totally agree)” → “7 (totally agree).” (scale anchors)

Response:

We greatly appreciate the reviewer's thorough reading and valuable suggestions. We have carefully addressed each of the mentioned points in the revised manuscript:

We have added “the” in the abstract accordingly.

Although previous studies often rely on simple correlation and regression analyses without comparing cognition across roles in the ultimatum game (UG) and the dictator game (DG). (Lines 14-16 on Page 1 of the revised manuscript)

We have removed “For” in Abstract, line 20:

Responders with fairness reputations… (Lines 20 on Page 1 of the revised manuscript)

The duplicated word in “shown in in Figure 2C” has been corrected to “shown in Figure 2C.”

The dictator’s decision-making procedure was shown in Figure 2C. (Lines 173 on Page 5 of the revised manuscript)

In Methods 2.3, line 190, “two-dimension social cognition” has been amended to “two-dimensional social cognition.”

Finally, we examined the relationship between categorical and two-dimensional social cognition within and between roles. (Lines 220-221 on Page 6 of the revised manuscript)

In Methods 2.2, line 165, the spacing in “7(totally agree)” has been adjusted to “7 (totally agree)” as recommended.

Afterward, participants were asked to evaluate their social perceptions of the players’ behavior on a seven-point Likert scale that range from 1 (totally disagree) to 7 (totally agree). (Lines 179-180 on Page 5 of the revised manuscript)

Reviewer 2 Report

Comments and Suggestions for Authors

This study explores how the fairness reputations of different roles (proposers, responders, dictators) in economic games influence observers' social cognitions. It applies a network analysis to examine the relationships between warmth-competence dimensions and categorical social cognitions. The authors use a convenience sample of university students in Hangzhou, through the university’s bulletin board system. After data examination and balancing participants with different social value orientations (SVOs), the authors included 122 valid participants in the study. The students are then asked to rate each role on categorical traits (fairness, trustworthiness, altruism, cooperation) and on warmth–competence dimensions. The study finds that warmth serves as a central node across all role networks, with distinct patterns for responders compared to proposers and dictators. Specifically, the network comparison results in lines 311-333 offer interesting findings. Responder networks show weaker associations between warmth-fairness and fairness-cooperation compared to proposer/dictator networks. Responder networks also show stronger associations between warmth-altruism and competence-fairness. Finally, responder networks indicate negative associations between fairness-altruism (which proposer/dictator networks lack). The ANOVA results for responders in lines 264-267 are also noteworthy, wherein the results show that fair responders have more positive perceptions (seen as fairer, trustworthy, and competent). Simultaneously, fair responders are seen as less altruistic, cooperative, and warm. It appears that responders show contradictory social cognitions wherein they're perceived as both more competent/fair but less warm/cooperative when they have fair reputations. This creates the "high competence-low warmth" profile. In contrast, proposers and dictators with fair reputations are consistently perceived more positively across all dimensions. In sum, fair proposers/dictators are judged more positively than unfair counterparts, and effects are larger for dictators than for proposers. Individualists discriminate reputations more strongly, while warmth is the most central node in all networks.

These findings are quite substantive. The topic is also of significant value, offering an analysis at the intersection of behavioral economics and social cognition. The study and the experimental design show some potential. However, several conceptual, design, and statistical issues need clarification or additional analyses.

Major comments:

  1. Methodology:
  • In this study, the fairness reputation manipulation is weakened by the inclusion of filler trials where “fair” players sometimes behave unfairly (e.g., a fair proposer making a 9:1 offer). This inconsistency risks reputation contamination, as participants may no longer perceive a stable fairness pattern but rather a mixed or strategic one. As a result, the manipulation may not cleanly distinguish between fair and unfair reputations, reducing construct validity and interpretability of the findings. Specifically, the inclusion of "filler trials" where fair players act unfairly (20% of trials) fundamentally compromises the reputation manipulation. If fair proposers make 9:1 offers in 2/10 trials, their reputation is no longer consistently "fair." This confounds the entire experimental design. One way to fix this would be to exclude filler trails from the reputational signal by recomputing each target’s reputation only on the 8/10 consistent trails and re-running the analyses using the cleaner reputation. This could be added as a robustness check. Another simpler way to do this would be to first define “fair” if ≥80% of observed actions meet the fairness rule; otherwise, “unfair.” Reclassify targets by this rule and re-estimate effects.
  • There is also a power analysis inconsistency in lines 109-111 and 186-188. The authors calculate n=108 needed for f=.25 effect sizes, but then report sensitivity analysis showing they can only detect f=.11-.13 effects. Given their reported effect sizes (e.g., ηp² = .05 for three-way interactions), key analyses may be underpowered. As a bare minimum, the authors need to correct or acknowledge the inconsistency in reporting and offer a clarification.
  • The authors run numerous separate ANOVAs on highly correlated outcomes (fairness, trustworthiness, altruism, cooperation, warmth, competence). This can inflate Type I error. To address these issues, the authors can either use Generalized Linear Mixed models with random intercepts for participants and stimuli/role, and multivariate approaches (MANOVA or, better, a multilevel SEM) to model outcome covariance. If the authors want to keep ANOVAs, they can implement family-wise error control (e.g., Holm) across the family of outcomes per hypothesis.
  • The interpretation of statistical results in lines 241-242 and 251-252 can be improved to avoid cherry-picking a few results. For transparency, primary results should be summarized in the main text. As a bare minimum, include the means and SD/SE of offers by “fair” vs “unfair” proposers/dictators, and acceptance rates by “fair” vs “unfair” responders in the main tables. The three-way interaction (Reputation × Role × SVO) on fairness ratings should be described explicitly, with simple-simple effects and CIs. Consider reporting effect sizes and confidence intervals in the main text.
  • In the paper (lines 218–224), the authors admit that the assumption of equal variances (homogeneity of variance) was violated for warmth ratings. That’s important because ANOVA assumes equal variances across groups. But their reported “fix” is to use Greenhouse–Geisser corrections, which actually address a different assumption, sphericity in repeated measures, not heterogeneity of variance. In other words, they applied the wrong correction for the problem they described. A correct way of handling heterogeneity would be to use methods like Welch’s ANOVA, robust standard errors, or transformations. Greenhouse–Geisser, doesn’t solve this issue.

  1. Interpretation of the results
  • In lines 416-427, the authors explain the SVO × reputation effects by appealing to motivational attributions (e.g., individualists infer strategy vs altruism), but they did not measure motive attributions directly. That means their interpretation is speculative, not empirically supported, and risks circular reasoning. Highlighting this is important.
  • The study only uses Chinese university students, and fairness/cooperation norms are known to vary by culture and age group. This is a genuine external validity limitation, especially for a paper making claims about “social cognition” in general. The authors do acknowledge this, but only briefly in their Limitations section (Lines 441–443). They write that because the sample consisted solely of Chinese university students, the results may not generalize across cultural contexts and should be replicated in more diverse populations. My concern is not just generalizability but also whether the “competent–cold” responder pattern might be culture-specific. The authors could acknowledge this limitation given that the well-documented cultural differences in fairness norms, it is unclear whether the competent–cold pattern for fair responders would replicate outside a Chinese student sample.

Minor comments:

  • For proposers/dictators, “fairness” is defined via offer distributions. On the other hand, for responders, it’s defined by rejecting unfair offers. These operationalizations imply different behavioral costs, motives, and informational bases, making between-role comparisons nontrivial. It’s crucial to explicitly justify that these role-specific definitions identify the same latent construct (“fairness reputation”) rather than role-specific norms (e.g., altruistic giving vs altruistic punishment). Please report the exact thresholds used to classify fair vs unfair behavior that participants observed (you list example sequences, but not the rule participants were expected to infer). Finally, it will be useful to discuss whether observed effects reflect normative expectations (equality, joint payoff maximization, reciprocity) rather than “fairness” per se, and consider relabeling constructs or adding a conceptual figure.
  • In lines 112-115, the authors write that “A total of 170 participants took part in this experiment. After data examination 112 and balancing the number of participants with different SVOs (see Supplementary Mate- 113 rials for details), 122 valid participants were included in the data analyses (61 males, Mage 114 = 20.68 years, SE = 1.94, ranging from 18 to 29 years).” This sounds like post-hoc selective retention that can bias estimates. It’s important to specify whether the authors use a priori exclusion criteria as well. Clarify the screening/exclusion process. How many participants were excluded, and for what specific reasons (e.g., failed comprehension check, incomplete data)?

General notes:

  • The authors should consider reporting the exact means and 95% confidence intervals on the bar/line plots rather than relying primarily on significance stars. This will give readers a clearer sense of effect size and precision.
  • The authors can also consider using standardized figure captions so that all abbreviations (FP, UP, FR, etc.) are defined on first use. Right now, some are only explained in-text.
  • The authors should also provide more detail on randomization. Was the trial order randomized within participants? Were the six role/reputation conditions counterbalanced?

Language and typos (minor):

  • There are a few awkward and redundant phrases which can be easily improved. For example, “as player of the corresponding roles” (line 127) is very confusing. It should be “as players in the corresponding roles”. So, the sentence reads better as “Then, participants engaged in decision-making processes as players in the corresponding roles, with the aim of enhancing their comprehension of the established rules.”
  • There is a typo in line 158. “in in Figure”.
  • The sub-heading 4.1 “The influence of fairness reputation on social cognition to proposes and dictators” should be “The influence of fairness reputation on social cognition to proposers and dictators”

Author Response

Dear Editor,

Manuscript Number: behavsci-3882873

Title: Warmth centrality in social cognitive networks of fairness reputation across players in the Ultimatum and Dictator Games

Thank you very much for providing me the opportunity to revise my manuscript. On behalf of my co-authors, I enclose a revised version of the above paper for submission to Behavioral Sciences. The following notes detail the changes made to the revised manuscript in response to the comments.

Each comment is presented, followed by a response.

Reviewer #2:

This study explores how the fairness reputations of different roles (proposers, responders, dictators) in economic games influence observers' social cognitions. It applies a network analysis to examine the relationships between warmth-competence dimensions and categorical social cognitions. The authors use a convenience sample of university students in Hangzhou, through the university’s bulletin board system. After data examination and balancing participants with different social value orientations (SVOs), the authors included 122 valid participants in the study. The students are then asked to rate each role on categorical traits (fairness, trustworthiness, altruism, cooperation) and on warmth–competence dimensions. The study finds that warmth serves as a central node across all role networks, with distinct patterns for responders compared to proposers and dictators. Specifically, the network comparison results in lines 311-333 offer interesting findings. Responder networks show weaker associations between warmth-fairness and fairness-cooperation compared to proposer/dictator networks. Responder networks also show stronger associations between warmth-altruism and competence-fairness. Finally, responder networks indicate negative associations between fairness-altruism (which proposer/dictator networks lack). The ANOVA results for responders in lines 264-267 are also noteworthy, wherein the results show that fair responders have more positive perceptions (seen as fairer, trustworthy, and competent). Simultaneously, fair responders are seen as less altruistic, cooperative, and warm. It appears that responders show contradictory social cognitions wherein they're perceived as both more competent/fair but less warm/cooperative when they have fair reputations. This creates the "high competence-low warmth" profile. In contrast, proposers and dictators with fair reputations are consistently perceived more positively across all dimensions. In sum, fair proposers/dictators are judged more positively than unfair counterparts, and effects are larger for dictators than for proposers. Individualists discriminate reputations more strongly, while warmth is the most central node in all networks.

These findings are quite substantive. The topic is also of significant value, offering an analysis at the intersection of behavioral economics and social cognition. The study and the experimental design show some potential. However, several conceptual, design, and statistical issues need clarification or additional analyses.

Major comments:

Q1: Methodology

In this study, the fairness reputation manipulation is weakened by the inclusion of filler trials where “fair” players sometimes behave unfairly (e.g., a fair proposer making a 9:1 offer). This inconsistency risks reputation contamination, as participants may no longer perceive a stable fairness pattern but rather a mixed or strategic one. As a result, the manipulation may not cleanly distinguish between fair and unfair reputations, reducing construct validity and interpretability of the findings. Specifically, the inclusion of "filler trials" where fair players act unfairly (20% of trials) fundamentally compromises the reputation manipulation. If fair proposers make 9:1 offers in 2/10 trials, their reputation is no longer consistently "fair." This confounds the entire experimental design. One way to fix this would be to exclude filler trails from the reputational signal by recomputing each target’s reputation only on the 8/10 consistent trails and re-running the analyses using the cleaner reputation. This could be added as a robustness check. Another simpler way to do this would be to first define “fair” if ≥80% of observed actions meet the fairness rule; otherwise, “unfair.” Reclassify targets by this rule and re-estimate effects.

There is also a power analysis inconsistency in lines 109-111 and 186-188. The authors calculate n=108 needed for f=.25 effect sizes, but then report sensitivity analysis showing they can only detect f=.11-.13 effects. Given their reported effect sizes (e.g., ηp² = .05 for three-way interactions), key analyses may be underpowered. As a bare minimum, the authors need to correct or acknowledge the inconsistency in reporting and offer a clarification.

The authors run numerous separate ANOVAs on highly correlated outcomes (fairness, trustworthiness, altruism, cooperation, warmth, competence). This can inflate Type I error. To address these issues, the authors can either use Generalized Linear Mixed models with random intercepts for participants and stimuli/role, and multivariate approaches (MANOVA or, better, a multilevel SEM) to model outcome covariance. If the authors want to keep ANOVAs, they can implement family-wise error control (e.g., Holm) across the family of outcomes per hypothesis.

The interpretation of statistical results in lines 241-242 and 251-252 can be improved to avoid cherry-picking a few results. For transparency, primary results should be summarized in the main text. As a bare minimum, include the means and SD/SE of offers by “fair” vs “unfair” proposers/dictators, and acceptance rates by “fair” vs “unfair” responders in the main tables. The three-way interaction (Reputation × Role × SVO) on fairness ratings should be described explicitly, with simple-simple effects and CIs. Consider reporting effect sizes and confidence intervals in the main text.

In the paper (lines 218–224), the authors admit that the assumption of equal variances (homogeneity of variance) was violated for warmth ratings. That’s important because ANOVA assumes equal variances across groups. But their reported “fix” is to use Greenhouse–Geisser corrections, which actually address a different assumption, sphericity in repeated measures, not heterogeneity of variance. In other words, they applied the wrong correction for the problem they described. A correct way of handling heterogeneity would be to use methods like Welch’s ANOVA, robust standard errors, or transformations. Greenhouse–Geisser, doesn’t solve this issue.

Response:

Validity of Fairness Reputation Manipulation

We sincerely appreciate the reviewer for this valuable comment and apologize for not having clearly explained the rationale for the filler trials.

In our experimental design, each role’s decision sequence consisted of ten trials to present a stable behavioral pattern of fairness or unfairness. Fair proposers and dictators offered fair allocations (5:5 or 6:4) in eight out of ten trials and unfair allocations (9:1 or 8:2) in two trials. Conversely, unfair proposers and dictators made eight unfair offers (9:1 or 8:2) and two fair ones (5:5 or 6:4). In the responder observation trials, unfair offers (9:1 or 8:2) appeared eight times and fair offers (5:5 or 6:4) twice. Responders with fair reputations typically rejected unfair offers and accepted fair ones, whereas unfair responders accepted both types of offers. This deterministic behavioral pattern, combined with filler trials, was designed to induce a clear and consistent fairness reputation while maintaining ecological plausibility and reducing participants’ suspicion of the manipulation (Osinsky et al., 2014). Moreover, previous research has shown that in ultimatum games, the acceptance rate for highly unfair offers such as 8:2 or 9:1 is not zero but around 20% (Güth et al., 1982), supporting the realism of including such filler behaviors. This deterministic behavioral pattern, combined with filler trials, was designed to induce a clear and consistent fairness reputation while maintaining ecological plausibility and reducing participants’ suspicion of the manipulation (Osinsky et al., 2014). Furthermore, previous research shows that for 8:2 or 9:1 split, acceptance rates are not zero but around 20% (Güth et al., 1982), which informed our design.

We added an operational definition of fair and unfair reputation in the Methods section 2.2:

A player was classified as “fair” if ≥80% of their observed actions met the fairness rule, and classified as “unfair” if ≥80% of their observed actions met the unfairness rule. (Lines 143-145 on Page 4 of the revised manuscript)

We also clarified that each participant observed the complete ten-round behavioral sequence of one role before making social-cognitive evaluations. The order of allocations was fixed across participants to ensure identical observation conditions. Moreover, the presentation of fair and unfair reputations was counterbalanced between participants: half of them viewed the fair target first, followed by the unfair target, and the other half viewed them in the opposite order.

Given this design, it was not possible to empirically isolate the specific effect of the filler trials themselves on the attenuation or polarization of fairness judgments, as all participants were exposed to the same mixed sequences. We have therefore acknowledged this as a methodological limitation, suggesting that future studies could employ behavioral sequences without fillers to establish a purer reputation manipulation. This would allow for a direct investigation into how the presence of filler trials modulates perceived reputation strength.

Fourth, we included a small number of inconsistent trials to enhance the ecological validity of the task and to prevent participants from perceiving the fairness-reputation manipulation as overly artificial or deterministic. However, because these fillers intentionally deviated from the assigned fairness reputation, they may have introduced ambiguity regarding the strength or consistency of each identity’s reputation. Although the presentation order was fixed within each role and the assignment of fair and unfair reputations was counterbalanced across participants, the present design does not allow us to determine whether the inclusion of fillers attenuated or polarized participants’ fairness judgments. Future studies could employ stimulus materials without such fillers to establish a more distinct and uncontaminated manipulation of fairness reputation and to assess the potential impact of inconsistent behavioral cues on reputation perception. (Lines 535-545 on Page 19 of the revised manuscript)

In addition, we have added the following statement to the Limitations section to address this issue of ecological validity:

Moreover, although the deterministic behavioral sequences and filler trials followed established paradigms in social decision-making research to ensure consistent reputation cues, such fixed patterns may have increased the salience of fairness cues and reduced ecological validity. Future studies should employ more dynamic, probabilistic, and interactive designs to better approximate real-world reputation formation. (Lines 520-525 on Page 19 of the revised manuscript)

References:

Güth, W., Schmittberger, R., & Schwarze, B. (1982). An experimental analysis of ultimatum bargaining. Journal of Economic Behavior and Organization, 3(4), 367–388. https://doi.org/10.1016/0167-2681(82)90011-7

Osinsky, R., Mussel, P., Öhrlein, L., & Hewig, J. (2014). A neural signature of the creation of social evaluation. Social Cognitive and Affective Neuroscience, 9(6), 731–736. https://doi.org/10.1093/scan/nst051

Power analysis inconsistency

We thank the reviewer for raising this important point regarding the consistency of our power analysis. We apologize for the lack of clarity in our initial description. The two power analyses we reported served different purposes:

  1. A priori power analysis (conducted before data collection) was used to determine the minimum sample size required to detect a medium effect size (f = .25), as conventionally defined by Cohen (1988). This yielded the optimal sample size of 108.
  2. Sensitivity power analysis (conducted after data collection) was used to determine the minimum effect size actually detectable with our final dataset (N = 122). This analysis indicated detectable effects of f = .13 for key mixed-ANOVA tests.

We understand the reviewer’s concern that our key analyses might be underpowered given the sensitivity result. We wish to reassure the reviewer on this point by comparing the detected effect sizes in our study to our achieved sensitivity.

The three-way interaction with ηp² = .05 corresponds to an effect size of f ≈ 0.23. This effect (f = .23) is substantially larger than the minimum effect (f = .13) our study could detect. A post-hoc power analysis for an effect of f = .23 with our sample size (N = 122) indicated that power for this specific effect was approximately 1-β=.0.99, which provides a good degree of confidence in this result. While it is true that our study was not powered to detect very small effects (f < 0.13), the key effects we reported and interpreted are all above this threshold. We have now added the following clarification to the manuscript to ensure full transparency:

Sensitivity power analysis, with N = 122, α = .05, power (1 - β) = .90, detected minimum effect sizes of f = .13 for the 2 × 2 × 2 mixed ANOVA and f = .16 for the 2 × 2 mixed ANOVA. The above details of the sensitivity power analysis are explicitly reported in the Supplementary Material. All reported significant effects exceeded this threshold, suggesting they were sufficiently powered. However, we acknowledge that the study may be underpowered to detect very small interaction effects. (Lines 206-211 on Page 6 of the revised manuscript)

Family-wise error control (Holm) in ANOVAs

We thank the reviewer for raising this important methodological concern regarding Type I error inflation from multiple correlated ANOVAs. In response to this valuable suggestion, we have implemented family-wise error control using Holm-Bonferroni corrections across the family of outcomes for each hypothesis. All statistical results in the manuscript now report Holm-Bonferroni corrected p-values. We have added the following revisions in the Methods section:

To control for Type I error inflation due to multiple correlated dependent variables, we applied Holm-Bonferroni corrections within logical hypothesis families. For each experimental hypothesis (e.g., main effects of fairness reputation, SVO, role, and their interactions), corrections were applied across the six dependent variables (fairness, trustworthiness, altruism, cooperation, warmth, and competence). All reported p-values reflect these conservative corrections. (Lines 214-219 on Page 6 of the revised manuscript)

For the proposers and dictators, a series of 2 (fairness reputation: fair vs. unfair) × 2 (SVO: prosocial vs. individualistic) × 2 (role: proposer vs. dictator) mixed-design ANOVAs across six dependent variables (fairness, trustworthiness, altruism, cooperation, warmth, and competence) were conducted. Each ANOVA included three main effects, three two-way interactions, and one three-way interaction, yielding a total of 42 hypothesis tests. To control the family-wise Type I error inflation caused by multiple outcomes, Holm corrections were applied within each family of effects.

After correction, only four results shifted from significant to non-significant: For fairness ratings: the fairness reputation × role interaction, and the SVO × fairness reputation × role three-way interaction; For cooperative ratings: the SVO × role interaction; For warmth ratings: the SVO × role interaction.

The uncorrected and Holm-adjusted p values for all tests for the proposers and dictators are presented in the table below:

Dependent variable

Effect

p.raw

p.holm

fairness

SVO

.818

1

fairness

fairness reputation

<.001

<.001

fairness

SVO × fairness reputation

.995

1

fairness

Role

<.001

<.001

fairness

SVO × Role

.003

.018

fairness

fairness reputation × Role

.021

.063

fairness

SVO × fairness reputation × Role

.016

.096

trustworthiness

SVO

.922

1

trustworthiness

fairness reputation

<.001

<.001

trustworthiness

SVO × fairness reputation

.832

1

trustworthiness

Role

.002

.002

trustworthiness

SVO × Role

.58

1

trustworthiness

fairness reputation × Role

<.001

<.001

trustworthiness

SVO × fairness reputation × Role

.639

1

altruism

SVO

.447

1

altruism

fairness reputation

<.001

<.001

altruism

SVO × fairness reputation

.913

1

altruism

Role

<.001

<.001

altruism

SVO × Role

.425

1

altruism

fairness reputation × Role

<.001

<.001

altruism

SVO × fairness reputation × Role

.938

1

cooperation

SVO

.807

1

cooperation

fairness reputation

<.001

<.001

cooperation

SVO × fairness reputation

.986

1

cooperation

Role

<.001

<.001

cooperation

SVO × Role

.023

.095

cooperation

fairness reputation × Role

.067

0.134

cooperation

SVO × fairness reputation × Role

.129

0.645

warmth

SVO

.885

1

warmth

fairness reputation

<.001

<.001

warmth

SVO × fairness reputation

.837

1

warmth

Role

<.001

<.001

warmth

SVO × Role

.029

.118

warmth

fairness reputation × Role

<.001

<.001

warmth

SVO × fairness reputation × Role

.911

1

competence

SVO

.994

1

competence

fairness reputation

.002

.002

competence

SVO × fairness reputation

.524

1

competence

Role

<.001

<.001

competence

SVO × Role

.606

1

competence

fairness reputation × Role

.089

.134

competence

SVO × fairness reputation × Role

.857

1

Based on the Holm-corrected results, we revised the descriptions in the Results section to ensure that all reported inferential conclusions reflect the adjusted statistical significance. In line with these revisions, we modified the Discussion section accordingly:

Our study found some evidence that individualists and prosocials may differ in their perception of fairness between proposers and dictators, though this pattern was observed specifically for fairness ratings after applying rigorous statistical corrections. One possible explanation for this pattern is that observers tend to assume that others share similar motivations, which may lead to differences in their perception and evaluation of others’ behaviors (Epley et al., 2004). Under this account, prosocials may be more inclined to assume that others also have prosocial motivations, while individualists may tend to assume that others are motivated by self-interest (Bogaert et al., 2008; Qi et al., 2017). This could lead individualists to infer that proposers’ fair distributions represent strategic behaviors to avoid rejection, while perceiving dictators’ fair distributions as potentially driven by genuine altruism. However, we stress that this interpretation remains speculative, as motive attributions were not directly measured in the present study. Future research should therefore incorporate explicit assessments of motive attributions or use modeling approaches to test whether reputation evaluations are indeed mediated by perceived intentions. (Lines 479-493 on Page 18 of the revised manuscript)

For the responder, a series of 2 (fairness reputation: fair vs. unfair) × 2 (SVO: prosocial vs. individualistic) mixed-design ANOVAs across six dependent variables (fairness, trustworthiness, altruism, cooperation, warmth, and competence) were conducted.

Each ANOVA included two main effects and one two-way interactions, yielding a total of 18 hypothesis tests. To control the family-wise Type I error inflation caused by multiple outcomes, Holm corrections were applied within each family of effects. The significance of all 18 p values remained unchanged after correction.

The uncorrected and Holm-adjusted p values for all tests for the responder are presented in the table below:

Dependent variable

Effect

p.raw

p.holm

fairness

SVO

.384

1

fairness

fairness reputation

<.001

<.001

fairness

SVO × fairness reputation

.964

1

reliable

SVO

.509

1

reliable

fairness reputation

.005

0.005

reliable

SVO × fairness reputation

.579

1

altruism

SVO

.253

1

altruism

fairness reputation

<.001

<.001

altruism

SVO × fairness reputation

.279

1

cooperation

SVO

.51

1

cooperation

fairness reputation

<.001

<.001

cooperation

SVO × fairness reputation

.243

1

warmth

SVO

.179

1

warmth

fairness reputation

<.001

<.001

warmth

SVO × fairness reputation

.664

1

competence

SVO

.898

1

competence

fairness reputation

<.001

<.001

competence

SVO × fairness reputation

.587

1

The implementation of these corrections has refined our conclusions, with only the most robust effects remaining statistically significant. We believe this approach strikes an appropriate balance between statistical rigor and theoretical interpretability while directly addressing the reviewer's valid concern about error inflation.

Statistical results of Figure 3 & 4

We thank the reviewer for this constructive comment to enhance the transparency and rigor of our statistical reporting. As recommended, we have now included two new tables (Table 1 and Table 2) in the main text. These tables summarize the means and standard errors (SE) for offers by “fair” vs. “unfair” proposers/dictators, as well as acceptance rates by “fair” vs. “unfair” responders.

Tables 1. The means and standard errors (SE) on social cognitions of “fair” vs. “unfair” proposers/dictators.

Social cognition

SVO

Role

Fairness Reputation

Mean (SE)

CI

Fairness

Prosocial

Proposer

Fair

4.17 (.16)

[3.86, 4.49]

Unfair

1.68 (.09)

[1.51, 1.85]

Dictator

Fair

4.49 (.17)

[4.16, 4.83]

Unfair

2.02 (.11)

[1.79, 2.24]

Individualistic

Proposer

Fair

3.81 (.16)

[3.49, 4.14]

Unfair

1.66 (.09)

[1.49, 1.84]

Dictator

Fair

4.92 (.17)

[4.57, 5.26]

Unfair

2.10 (.12)

[1.87, 2.34]

Trustworthiness

Prosocial

Proposer

Fair

3.95 (.17)

[3.62, 4.29]

Unfair

2.33 (.15)

[2.03, 2.64]

Dictator

Fair

4.60 (.16)

[4.28, 4.93]

Unfair

2.22 (.13)

[1.96, 2.49]

Individualistic

Proposer

Fair

3.93 (.17)

[3.59, 4.28]

Unfair

2.27 (.16)

[1.96, 2.59]

Dictator

Fair

4.61 (.17)

[4.27, 4.95]

Unfair

2.36 (.14)

[2.08, 2.63]

Altruism

Prosocial

Proposer

Fair

3.40 (.18)

[3.04, 3.75]

Unfair

2.05 (.20)

[1.65, 2.44]

Dictator

Fair

4.10 (.18)

[3.75, 4.44]

Unfair

1.89 (.16)

[1.56, 2.21]

Individualistic

Proposer

Fair

3.17 (.18)

[2.81, 3.53]

Unfair

1.86 (.21)

[1.46, 2.27]

Dictator

Fair

4.02 (.18)

[3.66, 4.38]

Unfair

1.83 (.17)

[1.49, 2.17]

Cooperation

Prosocial

Proposer

Fair

4.35 (.19)

[3.98, 4.72]

Unfair

1.81 (.09)

[1.62, 2.00]

Dictator

Fair

4.63 (.16)

[4.31, 4.96]

Unfair

2.05 (.12)

[1.81, 2.29]

Individualistic

Proposer

Fair

4.02 (.19)

[3.63, 4.40]

Unfair

1.71 (.10)

[1.52, 1.91]

Dictator

Fair

4.90 (.17)

[4.57, 5.23]

Unfair

2.08 (.13)

[1.83, 2.34]

Warmth

Prosocial

Proposer

Fair

3.81 (.16)

[3.50, 4.11]

Unfair

1.87 (.08)

[1.71, 2.03]

Dictator

Fair

4.49 (.17)

[4.15, 4.82]

Unfair

2.01 (.12)

[1.78, 2.24]

Individualistic

Proposer

Fair

3.58 (.16)

[3.26, 3.90]

Unfair

1.70 (.08)

[1.53, 1.86]

Dictator

Fair

4.64 (.17)

[4.30, 4.98]

Unfair

2.18 (.12)

[1.94, 2.42]

Competence

Prosocial

Proposer

Fair

4.06 (.13)

[3.80, 4.33]

Unfair

3.51 (.16)

[3.19, 3.83]

Dictator

Fair

4.32 (.13)

[4.07, 4.58]

Unfair

4.00 (.18)

[3.65, 4.35]

Individualistic

Proposer

Fair

3.94 (.14)

[3.66, 4.21]

Unfair

3.55 (.17)

[3.22, 3.89]

Dictator

Fair

4.31 (.13)

[4.04, 4.57]

Unfair

4.11 (.18)

[3.74, 4.47]

Tables 2. The means and standard errors (SE) on social cognitions of “fair” vs. “unfair” responders.

Social cognition

SVO

Fairness Reputation

Mean (SE)

CI

Fairness

Prosocial

Fair

4.83 (.20)

[4.42, 5.23]

Unfair

2.57 (.17)

[2.24, 2.90]

Individualistic

Fair

5.00 (.21)

[4.58, 5.42]

Unfair

2.73 (.17)

[2.39, 3.07]

Trustworthiness

Prosocial

Fair

4.29 (.18)

[3.93, 4.64]

Unfair

3.87 (.21)

[3.46, 4.28]

Individualistic

Fair

4.53 (.18)

[4.16, 4.89]

Unfair

3.92 (.21)

[3.49, 4.34]

Altruism

Prosocial

Fair

2.56 (.18)

[2.20, 2.91]

Unfair

5.35 (.21)

[4.93, 5.77]

Individualistic

Fair

3.00 (.19)

[2.63, 3.37]

Unfair

5.39 (.22)

[4.96, 5.82]

Cooperation

Prosocial

Fair

3.60 (.20)

[3.22, 3.99]

Unfair

5.13 (.20)

[4.73, 5.53]

Individualistic

Fair

3.97 (.20)

[3.57, 4.37]

Unfair

5.05 (.21)

[4.64, 5.47]

Warmth

Prosocial

Fair

3.62 (.17)

[3.28, 3.96]

Unfair

5.12 (.18)

[4.77, 5.47]

Individualistic

Fair

3.97 (.18)

[3.61, 4.32]

Unfair

5.35 (.18)

[4.98, 5.71]

Competence

Prosocial

Fair

4.35 (.16)

[4.04, 4.67]

Unfair

3.15 (.16)

[2.83, 3.46]

Individualistic

Fair

4.25 (.16)

[3.92, 4.57]

Unfair

3.21 (.16)

[2.89, 3.54]

However, after applying the appropriate statistical corrections, the three-way interaction (Reputation × Role × SVO) was no longer significant. Consequently, and in line with the principle of avoiding cherry-picking, we have removed the initial interpretation of this non-significant interaction from the Results section.

Heteroskedasticity and robustness

We thank the reviewer for this valuable suggestion regarding heteroskedasticity and robustness. In response, we have implemented Welch/Kenward-Roger corrections for our mixed-measures ANOVA analyses using the “mixed()” function from the “afex” package in R, which employs Kenward-Roger’s degrees of freedom approximation to provide robust inference when variance homogeneity assumptions are violated.

We can confirm that applying these robust corrections did not alter any of our substantive conclusions. All previously significant effects remained significant, and non-significant effects remained non-significant.

Specifically, for the proposers and dictators, the 2 (fairness reputation: fair vs. unfair) × 2 (SVO: prosocial vs. individualistic) × 2 (role: proposer vs. dictator) mixed-design ANOVA on warmth ratings revealed that the main effects of Role (F(1, 384.37) = 44.04, p < .001) and Fairness Reputation (F(1, 384.37) = 611.12, p < .001) remained significant. The Role × Type interaction (F(1, 384.37) = 9.92, p = .002) and the Role × Reputation interaction (F(1, 384.37) = 4.05, p = .044) also remained significant under the Kenward-Roger correction.

For the responder condition, the 2 (fairness reputation: fair vs. unfair) × 2 (SVO: prosocial vs. individualistic) mixed-design ANOVA on fairness ratings revealed that the main effect of Fairness Reputation remained significant, F(1, 240) = 141.75, p < .001. Similarly, the mixed-design ANOVA on competence ratings showed that the main effect of Fairness Reputation remained significant, F(1, 240) = 47.75, p < .001.

Following the reviewer’s suggestion, we have now transparently reported this methodological detail in the manuscript. Specifically, we have added the following footnote to the statistical analysis section:

Where applicable, we used Welch/Kenward-Roger’ corrections via the afex package (version 1.3-1) in R to ensure robust inference in the presence of potential heteroskedasticity. (Lines 212-213 on Page 6 of the revised manuscript)

All supplementary analysis scripts have been uploaded to our OSF project for transparency and reproducibility. We believe this addition strengthens the methodological rigor of our analysis and appreciate the reviewer’s guidance in this matter.

References:

Bogaert, S., Boone, C., & Declerck, C. (2008). Social value orientation and cooperation in social dilemmas: A review and conceptual model. British journal of social psychology, 47(3), 453-480. https://doi.org/10.1348/014466607X244970

Epley, N., Keysar, B., Van Boven, L., & Gilovich, T. (2004). Perspective taking as egocentric anchoring and adjustment. Journal of personality social psychology, 87(3), 327. https://doi.org/10.1037/0022-3514.87.3.327

Qi, Y., Wu, H., & Liu, X. (2017). The influences of social value orientation on prosocial behaviors: The evidences from behavioral and neuroimaging studies. Chinese Science Bulletin, 62(11), 1136-1144.

Q2: Interpretation of the results

In lines 416-427, the authors explain the SVO × reputation effects by appealing to motivational attributions (e.g., individualists infer strategy vs altruism), but they did not measure motive attributions directly. That means their interpretation is speculative, not empirically supported, and risks circular reasoning. Highlighting this is important.

The study only uses Chinese university students, and fairness/cooperation norms are known to vary by culture and age group. This is a genuine external validity limitation, especially for a paper making claims about “social cognition” in general. The authors do acknowledge this, but only briefly in their Limitations section (Lines 441–443). They write that because the sample consisted solely of Chinese university students, the results may not generalize across cultural contexts and should be replicated in more diverse populations. My concern is not just generalizability but also whether the “competent–cold” responder pattern might be culture-specific. The authors could acknowledge this limitation given that the well-documented cultural differences in fairness norms, it is unclear whether the competent–cold pattern for fair responders would replicate outside a Chinese student sample.

Response:

Interpretation of the SVO × reputation effects

We thank the reviewer for this insightful comment regarding the speculative nature of our motivational attribution interpretation. The reviewer is correct that we did not directly measure motive attributions, and we acknowledge that our interpretation, while theoretically grounded, remains inferential rather than empirically validated within the present data.

In response to this valuable feedback, we have revised the Discussion to explicitly acknowledge the speculative status of this interpretation and to reduce the degree of certainty in our claims. Specifically, we replaced definitive wording with more cautious phrasing (e.g., “may suggest,” “one possible explanation is”). We also clarified that our explanation represents a theoretically plausible interpretation rather than direct evidence of the underlying mechanism, and we emphasized that future research should incorporate explicit measures of motivational inferences to empirically test this proposed explanation. We have revised the statement in the Discussion section as follows:

Our study found some evidence that individualists and prosocials may differ in their perception of fairness between proposers and dictators, though this pattern was observed specifically for fairness ratings after applying rigorous statistical corrections. One possible explanation for this pattern is that observers tend to assume that others share similar motivations, which may lead to differences in their perception and evaluation of others’ behaviors (Epley et al., 2004). Under this account, prosocials may be more inclined to assume that others also have prosocial motivations, while individualists may tend to assume that others are motivated by self-interest (Bogaert et al., 2008; Qi et al., 2017). This could lead individualists to infer that proposers’ fair distributions represent strategic behaviors to avoid rejection, while perceiving dictators’ fair distributions as potentially driven by genuine altruism. However, we stress that this interpretation remains speculative, as motive attributions were not directly measured in the present study. Future research should therefore incorporate explicit assessments of motive attributions or use modeling approaches to test whether reputation evaluations are indeed mediated by perceived intentions. (Lines 479-493 on Page 18 of the revised manuscript)

Cultural differences

We sincerely appreciate the reviewer for highlighting the important issue of cultural and age-related differences in fairness and cooperation norms. In our study, which used Chinese university students, we observed that responders who rejected unfair offers were rated lower on warmth, reflecting a “competent-cold” pattern. Notably, this pattern has been documented in international and cross-cultural data as well as in individual stereotype research (Durante et al., 2017), suggesting that it may have some generalizability.

We have added a statement in the Limitations section to acknowledge this point and suggest that future studies examine:

Notably, the competent-cold pattern observed in fair responders in our study has also been documented in international and cross-cultural data, as well as in research on individual stereotypes (Durante et al., 2017). Future research could examine whether this competent-cold pattern for fair responders is observed across different cultural contexts. (Lines 509-513 on Page 19 of the revised manuscript)

References:

Bogaert, S., Boone, C., & Declerck, C. (2008). Social value orientation and cooperation in social dilemmas: A review and conceptual model. British journal of social psychology, 47(3), 453-480. https://doi.org/10.1348/014466607X244970

Durante, F., Tablante, C. B., & Fiske, S. T. (2017). Poor but warm, rich but cold (and competent): Social classes in the stereotype content model. Journal of Social Issues, 73(1), 138–157. https://doi.org/10.1111/josi.12208

Epley, N., Keysar, B., Van Boven, L., & Gilovich, T. (2004). Perspective taking as egocentric anchoring and adjustment. Journal of personality social psychology, 87(3), 327. https://doi.org/10.1037/0022-3514.87.3.327

Qi, Y., Wu, H., & Liu, X. (2017). The influences of social value orientation on prosocial behaviors: The evidences from behavioral and neuroimaging studies. Chinese Science Bulletin, 62(11), 1136-1144.

Minor comments:

Q3: For proposers/dictators, “fairness” is defined via offer distributions. On the other hand, for responders, it’s defined by rejecting unfair offers. These operationalizations imply different behavioral costs, motives, and informational bases, making between-role comparisons nontrivial. It’s crucial to explicitly justify that these role-specific definitions identify the same latent construct (“fairness reputation”) rather than role-specific norms (e.g., altruistic giving vs altruistic punishment). Please report the exact thresholds used to classify fair vs unfair behavior that participants observed (you list example sequences, but not the rule participants were expected to infer). Finally, it will be useful to discuss whether observed effects reflect normative expectations (equality, joint payoff maximization, reciprocity) rather than “fairness” per se, and consider relabeling constructs or adding a conceptual figure.

Response: We thank the reviewer for raising this important point regarding role-specific operationalizations of fairness. We acknowledge that for proposers and dictators, fairness was defined via offer distributions, whereas for responders it was defined by rejecting unfair offers. To ensure that both role-specific behaviors reflect the same latent construct, namely “fairness reputation”, we implemented clear and consistent classification rules, a target was classified as “fair” if ≥80% of observed actions met the fairness rule, and as “unfair” if ≥80% of observed actions met the unfairness rule. Participants were exposed to sequences consistent with these rules, allowing them to form stable fairness reputations across roles.

We also note that the observed effects may partly reflect normative expectations (e.g., equality, joint payoff maximization, or reciprocity) in addition to fairness (Rilling & Sanfey, 2011; Wang, Yang, Li, & Zhou, 2015). Specifically, the experimental paradigms we employed, including the Ultimatum Game and Dictator Game, involve multiple social norms such as fairness, altruism, reciprocity, and cooperation. Therefore, participants’ behaviors may be influenced by multiple motivations. Future research could further clarify how fairness reputation is formed across different roles using a conceptual figure and examine whether these social cognitive patterns generalize across cultures or populations.

We have supplemented this idea in the Limitations section:

Given that the present paradigms (the UG and DG) inherently involve fairness, altruism, and reciprocal cooperation, participants’ perceptions and evaluations may have been influenced by multiple overlapping social norms. Future studies could further disentangle these constructs by employing paradigms that isolate fairness from other cooperative motives and by testing whether similar cognitive patterns of fairness reputation emerge. (Lines 554-559 on Page 20 of the revised manuscript)

References:

Rilling, J. K., & Sanfey, A. G. (2011). The neuroscience of social decision-making. Annual Review of Psychology, 62, 23-48. https://doi.org/10.1146/annurev.psych.121208.131647

Wang, Y., Yang, L.-Q., Li, S., & Zhou, Y. (2015). Game theory paradigm: A new tool for investigating social dysfunction in major depressive disorders. Frontiers in Psychiatry, 6, 149463. https://doi.org/10.3389/fpsyt.2015.00128

Q4: In lines 112-115, the authors write that “A total of 170 participants took part in this experiment. After data examination 112 and balancing the number of participants with different SVOs (see Supplementary Mate- 113 rials for details), 122 valid participants were included in the data analyses (61 males, Mage 114 = 20.68 years, SE = 1.94, ranging from 18 to 29 years).” This sounds like post-hoc selective retention that can bias estimates. It’s important to specify whether the authors use a priori exclusion criteria as well. Clarify the screening/exclusion process. How many participants were excluded, and for what specific reasons (e.g., failed comprehension check, incomplete data)?

Response: We apologize for not providing a clear description of the participant exclusion and balancing procedures in the original manuscript, and thank the reviewer for pointing out this important issue.

A total of 170 participants completed the experiment. To ensure the reliability and accuracy of the results, we conducted a thorough data examination. Four participants with incorrect answers to the lie-detection item, four with survey completion times exceeding three standard deviations from the mean, and twelve with average estimates for role distribution per trial exceeding ±3 standard deviations (suggesting possible misunderstanding of role fairness reputation) were excluded, resulting in 150 participants entering further analysis. Before data analysis, 2 participants identified as altruistic or competitive types in the SVO task were excluded, because our focus was on the contrast between prosocial and individualistic orientations. This resulted in 59 individualistic and 89 prosocial participants. To balance group sizes and ensure comparable statistical power across SVO categories in the ANOVA, we retained 63 prosocial participants with the largest SVO angles, which better represented the prosocial tendency, while including all individualistic participants (n = 59). A few participants with identical boundary SVO angles were retained to avoid arbitrary exclusion.

As a result, 122 valid participants were included in the final analyses. No participants were excluded due to failed comprehension checks or incomplete data.

We have revised the Participant screening section in supplementary material to clarify this procedure as follows:

To balance the numbers of the prosocial and individualist, and given the substantial imbalance in group sizes (89 vs. 59), we balanced the two groups to ensure comparable statistical power in the ANOVA. Specifically, we retained 63 prosocial participants with the largest SVO angles, representing stronger prosocial tendencies, and included all individualistic participants (n = 59). Participants whose SVO angles fell on the boundary between groups were all retained to avoid arbitrary exclusion.

As a result, 122 valid participants (61 males, Mage = 20.68 years, SE = 1.94, range = 18–29 years) were included in the final analyses. All exclusions and balancing procedures were determined a priori to ensure data integrity and fair comparison between SVO categories. No participants were excluded due to failed comprehension checks or incomplete data. (Lines 14-23 on Page 1 of the revised Supplementary Material)

General notes:

Q5: The authors should consider reporting the exact means and 95% confidence intervals on the bar/line plots rather than relying primarily on significance stars. This will give readers a clearer sense of effect size and precision.

The authors can also consider using standardized figure captions so that all abbreviations (FP, UP, FR, etc.) are defined on first use. Right now, some are only explained in-text.

Response:

We are grateful to the reviewer for raising this important point regarding the presentation of our results. We fully acknowledge that supplementing significance stars with exact means and 95% confidence intervals provides a more complete statistical picture. Accordingly, we have incorporated these measures into the specified plots. Furthermore, we have standardized all figure captions and defined all abbreviations upon their first appearance.

Figure 3. The results of fairness reputation manipulation checks. (A) proposer role; (B) responder role; (C) dictator role. Significance level: *** p < .001. Error bars indicate standard errors.

Figure 4. Social cognition of proposers and dictators under different fairness reputation conditions. (A) categorical social cognition ratings: fairness, trustworthiness, altruism, and cooperation; (B) Two dimensions of social cognition ratings: warmth and competence. Significance level: * p < .05; ** p < .01; *** p < .001. Error lines indicate standard errors. Note. FP = Fair Proposer; UP = Unfair Proposer; FD = Fair Dictator; UD = Unfair Dictator.

Figure 5. Social cognition of responders under different fairness reputation conditions. (A) cate-gorical social cognition ratings: fairness, trustworthiness, altruism, and cooperation; (B) Two dimensions of social cognition ratings: warmth and competence. Significance level: * p < .05; ** p < .01; *** p < .001. Error lines indicate standard errors. Note. FR = Fair Responder; UR = Unfair Responder.

Q6: The authors should also provide more detail on randomization. Was the trial order randomized within participants? Were the six role/reputation conditions counterbalanced?

Response: We sincerely appreciate the reviewer for this valuable comment. In our experiment, the trial order within each role was fixed rather than randomized to ensure that participants could form a stable impression of each role’s behavioral pattern across trials. However, the reputation conditions were counterbalanced between participants: half of the participants first observed roles without fairness reputation and then those with fairness reputation, while the other half experienced the opposite order.

We have added this clarification to the 2.2 section of the revised manuscript:

The trial order within each role was fixed rather than randomized to ensure that participants could form a stable impression of each role’s behavioral pattern across trials. However, the reputation conditions were counterbalanced between participants: half of the participants first observed roles without fairness reputation and then those with fairness reputation, while the other half experienced the opposite order. (Lines 186-190 on Page 5 of the revised manuscript)

Q7: Language and typos (minor)

There are a few awkward and redundant phrases which can be easily improved. For example, “as player of the corresponding roles” (line 127) is very confusing. It should be “as players in the corresponding roles”. So, the sentence reads better as “Then, participants engaged in decision-making processes as players in the corresponding roles, with the aim of enhancing their comprehension of the established rules.”

There is a typo in line 158. “in in Figure”.

The sub-heading 4.1 “The influence of fairness reputation on social cognition to proposes and dictators” should be “The influence of fairness reputation on social cognition to proposers and dictators”

Response: We thank the reviewer for the helpful language suggestions. All issues have been corrected:

We have corrected it in the revised manuscript.

Then, participants engaged in decision-making processes as player in the corresponding roles… (Lines 139-140 on Page 4 of the revised manuscript)

The duplicated word in “shown in in Figure 2C” has been corrected to “shown in Figure 2C.”

The dictator’s decision-making procedure was shown in Figure 2C. (Lines 173 on Page 5 of the revised manuscript)

The sub-heading 4.1 was also revised:

4.1 The influence of fairness reputation on social cognition to proposers and dictators (Lines 410 on Page 17 of the revised manuscript)

Round 2

Reviewer 2 Report

Comments and Suggestions for Authors

The authors have addressed the concerns previously raised.